# Observations of traveling ionospheric disturbances driven by gravity waves from sources in the upper and lower atmosphere

Paul Prikryl[1], David R. Themens[1,2], Jaroslav Chum[3], Shibaji Chakraborty[4], Robert G. Gillies[5], James M. Weygand[6]

[1]Physics Department, University of New Brunswick, Fredericton, NB, Canada
[2]School of Engineering, University of Birmingham, Birmingham, UK
[3]Institute of Atmospheric Physics CAS, Prague, Czech Republic
[4]Space and Atmospheric Instrumentation Lab, Center for Space and Atmospheric Research, Daytona Beach, FL, USA
[5]Department of Physics and Astronomy, University of Calgary, Calgary, AB, Canada
[6]Earth, Planetary, and Space Sciences, University of California, Los Angeles, CA, USA

*Correspondence to*: Paul Prikryl (paul.prikryl@unb.ca)

## Abstract

The observed traveling ionospheric disturbances (TIDs) are attributed to atmospheric gravity waves generated by auroral sources in the lower thermosphere, or gravity waves generated in the troposphere by mid-latitude weather systems. TIDs are observed by the Poker Flat Incoherent Scatter Radar (PFISR), the Super Dual Auroral Radar Network (SuperDARN), the multipoint and multifrequency continuous Doppler sounders, and GNSS receivers using total electron content (TEC) mapping technique. At high latitudes, the solar wind-magnetosphere-ionosphere-thermosphere coupling modulated the dayside ionospheric convection and currents, the source of the equatorward propagating TIDs. The horizontal equivalent ionospheric currents are estimated from the ground-based magnetometer data using an inversion technique. At mid latitudes, the eastward to southeastward propagating medium-scale TIDs that were observed by the HF Doppler sounding system, as well as in the detrended TEC, are attributed to gravity waves that are generated by geostrophic adjustment processes and shear instability in the intensifying low-pressure systems and are identified in the stratosphere using the ERA5 meteorological reanalysis.

## 1. Introduction

The theory governing the propagation and effects of AGWs in the ionosphere was developed by Hines (1960) and their ionospheric sources have been recognized (Chimonas, 1970; Chimonas and Hines, 1970; Testud, 1970; Richmond, 1978). The relationship between atmospheric gravity waves (AGWs) and traveling ionospheric disturbances (TIDs) has been well established (Hocke and

Schlegel, 1996). Global propagation of medium- to large-scale GWs/TIDs has been linked to
auroral sources (Hunsucker, 1982; Hajkowicz, 1991; Lewis *et al.*, 1996; Balthazor and J., 1997).
The Worldwide Atmospheric Gravity-wave Studies (WAGS) program (Crowley and Williams,
1988; Williams *et al.*, 1993) showed that large-scale TIDs originate in auroral latitudes.
TIDs driven by AGWs that originate in the lower atmosphere come from a variety of sources,
including tropospheric weather systems (Bertin, Testud and Kersley, 1975; Bertin *et al.*, 1978;
Waldock and Jones, 1987; Nishioka *et al.*, 2013; Azeem *et al.*, 2015). Other known causes
AGWs/TIDs are solar flares (Zhang et al., 2019), total solar eclipses (Zhang *et al.*, 2017; Mrak *et*
*al.*, 2018), volcanic eruptions, earthquakes, and tsunamis (Yu, Wang and Hickey, 2017; Nishitani
*et al.*, 2019; Themens *et al.*, 2022).
Most of the early observations of TIDs were obtained by HF Doppler sounders, HF radars, and
incoherent scatter radars (Hunsucker 1982). More recently, the GNSS-derived total electron
content (TEC) measurements have become a commonly used technique to observe medium- to
large-scale TIDs (e.g., Afraimovich et al., 2000; Cherniak and Zakharenkova, 2018; Cheng et al.,
2021; Nykiel et al., 2024). Large-scale TIDs (LSTIDs) have wavelengths greater than 1000 km
and propagate at speeds 400-1000 m/s, while medium-scale TIDs (MSTIDs) have wavelengths of
several hundred kilometers and tend to propagate at speeds of 250–1000 m/s (Hunsucker, 1982).

van de Kamp et al. (2014) described two techniques to detect TIDs, one using the EISCAT
incoherent scatter radar near Tromsø, and the other using the detrended GPS TEC data. They
determined parameters characterizing TIDs and studied an event of January 20, 2010. While these
authors did not investigate the origin of the TIDs they suggested that the AGWs were most likely
generated at low atmospheric layers. Using the EISCAT Svalbard radars on February 13, 2001,
Cai et al. (2011) observed moderately large-scale TIDs propagating over the dayside polar cap that
were generated by the nightside auroral heating. It is noted that both these TID events occurred on
days following arrivals of corotating interaction regions (CIRs) at the leading edge of solar wind
high-speed streams that triggered moderate geomagnetic storms.

Frissell et al. (2016) concluded that polar atmospheric processes, namely the polar vortex, rather
than space weather activity are primarily responsible for controlling the occurrence of high-latitude
and midlatitude winter daytime medium-scale TIDs (MSTIDs). This paper has been frequently
cited to justify suggestions of polar vortex as a source of the observed MSTIDs, particularly when
geomagnetic activity is low. Recent papers (Becker et al., 2022; Bossert et al., 2022; Vadas et al.,
2023) discussed methods for assessing vortex generated GWs from model output. Vadas et al.
(2023) discussed observations of polar vortex generated GWs and subsequent secondary GW
generation in the polar region. Becker at al. (2022) performed simulations focusing on multi-step
vertical coupling by primary, secondary, and higher-order gravity waves of wintertime
thermospheric gravity waves and compared them with observed perturbations of total electron
content. They demonstrated that gravity waves generated from lower altitudes can propagate
equatorward. Bossert et al. (2022) discussed a strong TID/TAD event observed during a sudden
stratospheric warming (SSW) on January 18-19, 2013, and suggested that the large-scale
TIDs/TADs were related to geomagnetic activity despite low $Kp$ index. Thus, it is important to
continue discussing possible sources of GWs in the lower and upper atmosphere.

The solar wind coupling to the dayside magnetosphere (Dungey, 1961, 1995) generates variable
electric fields that map to the ionosphere driving the $\boldsymbol{E}\times\boldsymbol{B}$ ionospheric convection observed by
SuperDARN HF radars and ionospheric currents observed by network of ground-based
magnetometers. The mechanisms for energy transfer to the thermosphere can be Joule heating,
precipitation, or ion drag by Lorentz force (Richmond, 1979; Nishimura et al., 2020; Deng et al.,
2019). Nykiel et al. (2024) found that Joule heating is a primary energy source for the night-time
LSTIDs triggered in the auroral region, while the daytime LSTIDs can be also driven by
precipitating particles in the polar cusp. On the dayside, pulsed ionospheric flows (PIFs), that are
known to be associated with poleward moving auroral forms (precipitation), have been observed
to generate AGWs, which in turn modulate ionospheric densities resulting in MSTIDs propagating
equatorward observed by SuperDARN radars (Samson *et al*., 1989; 1990; Prikryl *et al*., 2005,

96    2022).


In this study, even in the case when geomagnetic activity is very low, PIFs are found to be the
source of AGWs/TIDs observed by PFISR and SuperDARN (Section 3.1). In other cases, the
AGWs/TIDs that were generated by solar wind Alfvén waves coupling to the dayside
magnetosphere are investigated in Section 3.2. The equatorward propagating TIDs were observed
in the detrended vertical vTEC and the radar ground-scatter power focused and defocused by TIDs
moving equatorward.
The tropospheric convection is often a source of gravity waves that propagate into the upper
atmosphere driving TIDs (e.g., Azeem, 2021; and references therein). In this study, gravity waves
generated in the troposphere by geostrophic adjustment processes and shear instability (Uccellini
and Koch, 1987) are considered to drive TIDs propagating eastward to southward that were
observed in the detrended vTEC maps and by the HF Doppler sounding system (Section 4). To
our knowledge, this physical mechanism has not been considered as a source of TIDs. The ERA5
meteorological reanalysis is used to show the gravity waves propagating into the stratosphere,
The aim of this study is to attribute the observed TIDs to sources in the upper (Section 3), and the
lower (Section 4) atmosphere. These observations show that AGWs provide both downward and
upward vertical coupling of the ionosphere and neutral atmosphere.

**2. Data sources and methods**
Fig. 1 shows a map of instruments used in this study as described below.

*2.1. Advanced Modular Incoherent Scatter Radar*
Data from the Advanced Modular Incoherent Scatter Radar (AMISR) located in Poker Flats, AK
(i.e., the Poker Flats Incoherent Scatter Radar - PFISR) was used in this study. AMISR radars use
phased-array beam-forming techniques on a pulse-to-pulse timescale to effectively sample various
look directions simultaneously (Heinselman and Nicolls (2008)).  These radars have been used to
investigate gravity wave propagation (Nicolls and Heinselman, 2007; Vadas and Nicolls, 2008).
PFISR is located at the Poker Flat Research Range (65.1°N, 147.5°W) near Fairbanks, Alaska
(Fig. 1a).  Due to its phased-array system, PFISR (like other AMISR systems) is able to operate in
a variety of different ionospheric sampling modes (often these modes can even be run
simultaneously by interleaving different pulse sequences).  During the time of interest for this
study (January 8-9, 2013), PFISR was running a 7-beam mode with interlaced Long Pulse (LP)
and Alternating Code (AC) modes.  Typically, the LP data is primarily used for sampling the F-
region, while the AC mode allows better resolution of E-region densities and parameters.  In this
study, the LP electron density data from beam 2 of this experiment (elevation 52.5º and azimuth -
7.8º) are used. The electron densities measured by an ISR such as PFISR are determined from the
total signal power returned from a given range gate. The LP mode in this experiment had a range
resolution of 25 km, resulting in an altitude resolution of 19.5 km for this beam.  To retrieve TIDs,
background densities are removed by applying Savitzky-Golay filter (Press and Teukolsky, 1990)
that has been used in previous studies (Zhang *et al.*, 2017; 2019). Other types of high-pass filters
and detrending methods would produce similar results.

*2.2. Multi-point and multi-frequency continuous HF Doppler sounding system*

The multi-point and multi-frequency continuous HF Doppler sounding system operating in the
Czech Republic is used to determine propagation velocities and azimuths of TIDs at the specific
heights of the signal reflections. The Doppler system consists of three transmitting sites Tx1, Tx2,
and Tx3 that are distributed in the western part of the Czech Republic (Tx1: 50.528°N, 14.567°E;
Tx2: 49.991°N, 14.538°E; Tx3: 50.648°N, 13.656°E) and receiver Rx located in Prague
(50.041°N, 14.477°E) (Fig. 1a). The system is operating at the frequencies of 3.59, 4.65 and 7.04
MHz. The reflection heights depend on the sounding frequency and change during the day and can
be obtained from ionograms measured by a nearby digital ionospheric sounder. The frequencies
of transmitters operating at a specific frequency are mutually shifted by about 4 Hz at different
transmitting sites. Thus, the signals of individual transmitters can be easily distinguished at the
receiving site. Locations of the reflection points are assumed in the middle between the individual
transmitter–receiver pairs, if projected to the ground. Horizontal distances between the individual
reflection points are about 30–50 km. Thus, the system is suitable for propagation analysis of
medium-scale TIDs.
The processing runs as follows. First, the time evolution of power spectral densities - Doppler
shift spectrograms are computed for each signal and the maximum of power spectral density
(characteristic Doppler shift) is found with selected time resolution suitable for the TIDs/GWs
analysis (30 or 60 s). The obtained time series then serve as an input for the propagation analysis
of TIDs (Chum et al., 2021). It should be noted that TID/GW cause movement of plasma and
therefore the Doppler shift. The propagation velocities and azimuths are then determined from the
time delays between the Doppler shifts recorded for different transmitter-receiver pairs and
expected distances of the reflection points in the ionosphere. Three different methods are used to
compute time delays between the observed signals (obtained from time series of the Doppler
shifts): (a) best beam slowness search, (b) unweighted and (c) weighted least square solution of
overdetermined set of equations, based on the time delays obtained for maxima of cross-correlation
functions between the individual signals. See Chum and Podolská (2018) for a more detailed
description and formulas. Mean values of the propagation velocities calculated from the time
delays obtained by these three different methods are presented further. At the same time,
uncertainties are estimated from the variance of the obtained results by these three methods as
standard deviations. Only velocities that satisfy the following criteria are considered (displayed):
the uncertainty of propagation velocity is less than 20% of its absolute value and the uncertainty
of azimuth is less than 10°. Analysis for signals Doppler shifts smaller than 0.1 Hz is also not
displayed.
A 2-D version of propagation analysis (in the horizontal plane) is applied here to analyse longer
time intervals (Chum et al. 2021). It should be noted that usable signals (Doppler shifts) are only
obtained if the signals reflect from the F2 layer. The Doppler shift of signal reflecting from the E
layer is usually negligible. The sounding frequency has also to be smaller than the critical
frequency of the ionosphere to receive the reflected signal.
*2.3. Super Dual Auroral Radar Network (SuperDARN)*SuperDARN constitutes a globally
distributed HF Doppler radar network, operational within the frequency range of 8 to 18 MHz,
encompassing both the northern and southern hemispheres across various latitudinal bands,
including middle, high, and polar zones. Each radar within this network measures the line-of-sight
(LoS) component of the drift velocity associated (Chisham et al., 2007; Nishitani et al., 2019). The
observations from SuperDARN encompass two principal forms of backscatter, namely, ground
and ionospheric scatters.  In the case of ground scatter, due to the high daytime vertical gradient
in the refractive index, the rays bend toward the ground and are reflected from surface roughness
and return to the radar following the same paths. This simulates a one-hop ground-to-ground
communication link that passes through the D-region 4 times. Ionospheric scatter is generated
when a transmitted signal is scattered from ionospheric irregularities. Typically, ground and
ionospheric scatters are associated with relatively lower & higher Doppler velocities and narrower
& wider spectral widths, respectively.

SuperDARN radars are phased-array systems with electronically steerable beams. These radars
use a narrow (~3.2°) azimuthal beam and a wide (~25-30°) vertical beam, typically employing an
array of 16-20 antennas. A standard 16-20 beam scan provides a field of view approximately 52-
62° wide in azimuth, extending from about 180 km to beyond 3000 km in range, with
measurements typically taken at 45 km intervals. The radars can operate across a wide range of
HF frequencies (8-20 MHz), allowing them to utilize various propagation modes and observe
diverse geophysical phenomena. They transmit multi-pulse sequences, and the resulting echoes
are processed to calculate multi-lag autocorrelation functions (ACFs). These ACFs, averaged over
3-6 seconds, are used to determine power, line-of-sight Doppler velocity, and Doppler spectral
width. The resulting fitted ACFs, known as Fitacf data, are generated from raw radar data using
version 2.5 of the FITACF algorithm (SuperDARN Data Analysis Working Group et al., 2022).

In this study, we use line-of-sight (LoS) Doppler velocities and ground scatter observations, i.e.,
fitacf2.5 datatsets, to characterize TIDs, with supplementary support from ionospheric convection
maps available at the SuperDARN Virginia Tech (VT) website (vt.superdarn.org) to validate their
sources. Prior research has demonstrated the utility of both scatter types in studies of pulsed
ionospheric flows (PIFs) (McWilliams, Yeoman and Provan, 2000; Prikryl et al., 2002) and TIDs
(Samson et al., 1990). MSTID signatures, embedded within radar backscatter, can be used to assess
TID phase and propagation direction (Frissell et al., 2014). Previous studies have relied on
fitacf2.5-derived parameters, like power and Doppler shift, for such analyses (e.g., Frissell et al.,
2014; Inchin et al., 2024). Fig. 1a displays fields-of-views and beams of the radars used in this
study.

*2.4. Global Navigation Satellite System (GNSS)*
The GNSS data for this were gathered from the same global networks of GNSS receivers used in
Themens et al. (2022), which constitute 5200-5800 stations, depending on the period. Examples
of the GNSS station distribution in the two local domains are shown in red dots in Figs. 1b and 1c.
Using the phase leveling and cycle slip correction method outlined by Themens et al. (2013),
whereby phase TEC is leveled to pseudorange TEC using an elevation-weighted mean, the LoS
total electron content (TEC) is determined from the differential phase and code measurements of
these systems. As detailed in Themens et al. (2015), the satellite biases are acquired from the
Center for Orbit Determination in Europe (CODE, ftp://ftp.aiub.unibe.ch/) and receiver biases are
determined using a simple Minimization of Standard Deviations approach, wherein a bias is
selected by iterating through test biases and minimizing the variance in vertical TEC (vTEC) from
all satellites in view over the course of a full day.

To characterize the TID structures using these data, LoS TEC measurements for each satellite-
receiver pair were detrended by first projecting the LoS TEC to vertical TEC (vTEC) using the
thin shell approximation at 350-km altitude and subtracting the sliding 60-minute average. More
details on this method can be found in Themens et al. (2022). The TEC anomalies are then binned
in 0.75-degree latitude and longitude bins for mapping.

It should be noted, that the detrending used in this analysis largely removes any impact of the
satellite biases and receiver biases on the detrended TEC, so the calculation of biases is largely not
needed. If we were concerned with timescales closer to the length of a full GNSS arc of lock,
residual bias effects on the projected vTEC may need to be considered but likely would not be
significant if at least an approximate bias is used.

*2.5. Spherical Elementary Current System (SECS)*
For this study we will be using the spherical elementary current system (SECS) method to calculate
a two-dimensional map of the ionospheric currents. Here we describe the SECS method. A two-
dimensional picture of the ionospheric currents can be derived from an array of well-spaced ground
magnetometers. The SECS method (Amm and Viljanen, 1999), which has been regularly applied
to the International Monitor for Auroral Geomagnetic Effects (IMAGE) ground magnetometer
array, can calculate the equivalent ionospheric currents (EICs), which are parallel to Earth's
surface, and the SEC amplitudes, which are a proxy for the field-aligned-currents. The SECS
technique defines two elementary current systems: a divergence-free elementary current system
with currents that flow entirely within the ionosphere and a curl-free current system whose
divergences represent the currents normal to the ionosphere. The superposition of these
divergence-free and curl-free current systems can reproduce a vector field on a sphere. For this
study we only calculate the divergence-free (equivalent ionospheric) currents, which are a
combination of the real Hall and Pedersen currents. The temporal and spatial resolutions of the
equivalent ionospheric currents are 10 s and 6.9° geographic longitude by 2.9° geographic latitude
(see, Amm et al., 2002 for more details). In general, the EICs are calculated from a matrix inversion
of ground magnetic disturbances. One of the important features of this technique is that it requires
no integration time of the magnetometer data. Unlike like AMPERE currents, which requires a 10
min integration time for many of the spacecraft to cross the polar region or the SWARM currents
that only provide a slice of the ionospheric currents as they cross the auroral region. The technique
(see, Weygand et al. 2011; their Figure 1) has been applied to magnetometers located in North
America and Greenland for the last 15+ years (Weygand et al. 2012; 2016).

In the European sector, ground magnetic perturbances due to the ionospheric currents and 1D
equivalent currents estimates are provided by the International Monitor for Auroral Geomagnetic
Effects (IMAGE) array of magnetometers.

*2.6. S*olar wind data
The solar wind data are provided by the OMNIWeb (http://omniweb.gsfc.nasa.gov) (King and
Papitashvili, 2005) and the Goddard Space Flight Center Space Physics Data Facility
(https://spdf.gsfc.nasa.gov/index.html). The magnetic field, proton density and velociy
measurements obtained by Advanced Composition Explorer (ACE) (Smith *et al.*, 1998) are used.

*2.7. ERA5 meteorological reanalysis*
The hourly reanalysis dataset ERA5 with spatial resolution of 0.25 x 0.25 degree (Hersbach et al.,
2020) is a product of the European Centre for Medium-Range Weather Forecasts (ECMWF). The
300-hPa geopotential height, horizontal winds at 300 hPa, and the divergence of the horizontal
wind at 150-hPa level are used. The alternating bands of convergence and divergence have been
interpreted by Plougonven and Teitelbaum (2003) as gravity waves propagating to the lower
stratosphere.

**3. AGWs/TIDs originating from lower thermosphere at high latitudes**

Solar wind high-speed streams (HSSs) are associated with high-intensity, long-duration
continuous auroral electrojet activity (HILDCAAs) that includes auroral substorms (Tsurutani and
Gonzalez, 1987; Tsurutani et al., 1990, 1995). HILDCAAs are caused by trains of solar wind
Alfvén waves (Belcher and Davis, 1971) that couple to the magnetosphere-ionosphere system
(Dungey, 1961, 1995). This coupling extends to the neutral atmosphere and ionosphere because it
is a source of aurorally excited gravity waves. Solar wind modulation of cusp particle signatures
was associated with ionospheric flows (Rae et al., 2004). Solar wind Alfvén waves can modulate
ionospheric convection and currents, most directly on the dayside, producing polar cap density
patches and AGW/TIDs (Prikryl *et al.*, 1999, 2005, 2022).

We now investigate cases of TIDs observed at high latitudes using the PFISR, SuperDARN and
GNSS data supported by solar wind and ground-based magnetometers. In Section 3.1, we examine
one case when PFISR in Poker Flat, Alaska, observed TIDs during geomagnetically quiet period
on January 8, 2013 in the context of solar wind coupling to show evidence that the observed TIDs
originated in the high-latitude dayside ionosphere poleward of Alaska, where there are no ground-
based magnetometers to sense the ionospheric currents, but ionospheric convection is observed by
SuperDARN. In Section 3.2, we present more examples of TIDs generated by solar wind-M-I-T
coupling on the dayside. The TIDs were observed in the North America and Europe by
SuperDARN and in the detrended vTEC.

*3.1. Event of January 8/9, 2013*
In the period from January 8 to 15 the PFISR beams scanned electron densities, $N_e$ (cm$^{-3}$), at
altitudes from 150 to 500 km. In the detrended TEC maps over Alaska ([https://aer-nc-](https://aer-nc-web.nict.go.jp/GPS/GLOBAL/MAP/2013/008/index.html)
[web.nict.go.jp/GPS/GLOBAL/MAP/2013/008/index.html](https://aer-nc-web.nict.go.jp/GPS/GLOBAL/MAP/2013/008/index.html)) the equatorward propagating TIDs
were observed on each day during the daytime hours when the PFISR density data show signatures
of downward propagating phase of TIDs. Fig. 2a shows $N_e$ in logarithmic scale as a function of
altitude observed by the radar beam 2 at temporal resolution of 3 min between 18:00 and 03:00
UT (09:00 and 18:00 LT) on January 8-9, 2013. The downward propagating phase of TIDs is
readily seen superposed on the background of high daytime densities. To remove the background
and highlight the TIDs with periods > 40 min the time series for each altitude are detrended using
a 33-point wide Savitzky-Golay filter (4$^{th}$ degree, 2$^{nd}$ order) (Fig. 2b). To show the equatorward
propagation of the TIDs across Alaska, Fig. 2c shows the detrended GNSS vTEC mapped at
latitude bins along the longitude of the PFISR.

The geomagnetic activity on January 8 was low, with the $Kp$-index $\leq 1$ except for a peak of 3- in
the last 3-hourly interval caused by a substorm that occurred in the European sector. The
northernmost magnetometer in Alaska in Barrow observed the north-south $X$-component magnetic
field perturbation of ~230 nT at 17:10 UT (see Fig. S1 in the Supplement) indicating the westward
electrojet. At this time, the IMF was pointing dawnward ($B_y < 0$) and eastward flows (see Fig. S2a
in the Supplement) in the dawn convection cell corresponded with the westward electrojet sensed
in Barrow. After 18:00 UT, as the IMF $B_y$ reversed to duskward (Figure 2c), the convection cells
receded further poleward of Alaska and the convection pattern become dominated by the dusk cell
(see Fig. S2b in the Supplement). At this time, the distant westward electrojet over Beaufort Sea
could no longer be detected by magnetometers.

The King Salmon Radar (KSR) beam 9 pointing northwest over the East Siberian Sea observed
positive (towards the radar) line-of-sight (LoS) velocities indicating quasiperiodic (20-50 min)
pulsed ionospheric flows (PIFs; Fig. 3a) in the dawn convection cell. At near ranges, the KSR
radar observed enhancements in the sea scatter power (Fig. 3b) caused by a series of equatorward
propagating TIDs. The Prince George Radar (PGR) beam 1 also observed the TIDs in the ground
scatter power with periods similar to those of PIFs and the TIDs observed by PFISR (Fig. 2). To
show the actual TID location in the ionosphere, instead of the slant range the ground-scatter range
mapping (Bristow, Greenwald and Samson, 1994; Frissell *et al.*, 2014) is applied in Fig. 3d.
Projected to the direction of beam 1, the estimated TIDs wavelengths ranged between ~200 and
500 km, and phase velocities between ~100 and 300 m/s.

The IMF southward turnings are expected to modulate the reconnection rate at the magnetopause
leading to intensifications of the ionospheric convection/currents in the cusp footprint. One of the
convection enhancements can be viewed in Fig. S2b in the Supplement. Pulsed ionospheric flows
are known to be associated with particle precipitation in poleward moving auroral forms (Rae et
al., 2004). In the cusp, particle precipitation and Joule heating can be comparable energy source
for AGWs/TIDs (Nykiel et al., 2024). The time series of the ACE IMF $B_y$ and $B_z$, as well as the
clock angle ($B_z$, $B_y$) counted from the geomagnetic north, with the 180° (dotted line) indicating
southward turnings of the IMF are shown time-shifted in Fig. 3c. Normalized FFT spectra of the
detrended IMF $B_z$, $B_y$, and the Prince George radar ground scatter power (beam 1, gate 30, mapped
GS range 730 km) are shown in the inset in Figure 3d. The spectra of the IMF $B_z$ and the Prince
George radar ground scatter power are very similar thus providing evidence that the generation of
TIDs was driven by solar wind coupling to the dayside magnetosphere. The clock angle controls
the reconnection rate at the magnetopause (Milan et al., 2012). The TIDs can be approximately
associated with southward IMF turnings (positive deflections of the clock angle values towards
180° marked by arrows in Fig. 3). Of course, this is an approximate correspondence. The IMF
observed by ACE does not represent exactly the IMF impacting the dayside magnetopause. It
would require at least a spacecraft in front of the bow shock to monitor the IMF (e.g., Prikryl et al.
2002). But the observations of pulsed ionospheric flows (PIFs) and corresponding TIDs points to
sources of these TIDs in the high-latitude ionosphere poleward of Alaska. Prikryl et al. (2005; their
Figure 2) showed that the observed TID propagation times, horizontal wavelengths and phase
velocities deduced from the ground-scatter are consistent with the results of ray tracing of the
gravity wave energy from an ionospheric source.

While we focused here on January 8/9, on each day during the PFISR experiment from January 8
to 15 the solar wind-MIT coupling that modulated PIFs in the ionospheric cusp footprint poleward
of Alaska launched TIDs that were observed by PFISR, as well as in the GNSS vTEC maps.
Similarly, in the European sector, dayside TIDs propagating equatorward from their sources in the
cusp over Svalbard were also observed. This can be viewed in Fig. S3 in the Supplement, and it is
consistent with previous studies of PIFs in the ionospheric cusp resulting in TIDs (Prikryl et al.,

372    2005; 2022).


*3.2. Events of November 1 and 4-5, 2014*

Fig. 4 shows the solar wind data indicating arrivals of HSS/CIRs marked by asterisks. The second
one triggered a minor geomagnetic storm with the *Dst* index (Gonzalez *et al.*, 1994) reaching
maximum negative value of −44 nT on November 4. Solar wind Alfvén waves permeate HSSs,
and along with CIRs, are highly geoeffective when IMF $B_z < 0$ (Tsurutani et al., 1987, 1995, 2006).
On November 1 and 5, 2014, the solar wind Alfvén waves are characterized by the Walén relation
between velocity $V$ and magnetic field $B$ (Yang, Chao and Lee, 2020). The components of the
corresponding components of the magnetic field ($B_y$ and $B_z$) and velocity ($V_y$ and $V_z$) observed by
ACE are correlated (Fig. 5a), which is a signature of solar wind Alfvén waves. For November 5,
this can be viewed in Fig. S4a in the Supplement.

In the European sector, the SuperDARN Hankasalmi radar observed PIFs in the cusp over
Svalbard, and equatorward propagating TIDs, which were also observed in the detrended vTEC.
Figs. 6 and 7 show the ionospheric LoS velocities and the radar scatter power (ground scatter
shown in grey color in the velocity plot) observed by the radar beam 11 on November 1 and 5,
respectively. The radar ground scatter (Figs. 6b and 7b) at ranges between 1000 and 1800 km
shows tilted bands due to equatorward propagating TIDs. Applying the ground scatter mapping
method (as already discussed in Section 3.1), the estimated MSTID wavelengths ranged between
~100 and 500 km, and phase velocities between ~100 and 300 m/s (Figs. S5a and S5b in the
Supplement).

The ground magnetic field perturbations of the X-component observed in Ny Ålesund (NAL;
https://space.fmi.fi/image/www/index.php) and 1D equivalent currents estimates that use all
IMAGE magnetometers are superposed. The radar observed a series of intensifications of the
negative (away from the radar) LoS velocities (PIFs) at ranges greater than ~2000 km on the
dayside, starting at ~07:00 UT with the onset of ionospheric currents fluctuations sensed by the
NAL magnetometer. The solar wind Alfvén waves modulated the dayside ionospheric currents
launching AGWs driving the equatorward propagating TIDs observed in the radar ground scatter
at ranges below ~2000 km. For November 1, Figs. 5b and 5c show the FFT spectra of detrended
time series of IMF $B_z$, solar velocity $V_z$, the NAL X-component, and the Hankasalmi radar ground
scatter power (beam 11, gate 25, slant range 1305 km) displaying peaks at similar
frequencies/periods. For November 5, the FFT spectra can be viewed in Figs. S4b and S4c in the
Supplement.

Figs. 8a and 8b show the TIDs observed in the detrended vTEC as alternating positive and negative
anomalies mapped along longitude of 15°E on November 1 and 5, respectively. The equatorward
propagating TIDs were observed at mid latitudes, but it is difficult to estimate their propagation
speeds as they appear to have been disrupted due to interference with TIDs from tropospheric
sources moving eastward to southeastward that are discussed in Section 4.

The arrival of the HSS/CIR on November 4 triggered a minor geomagnetic storm. Similar to cases
reported previously (Prikryl et al., 2022), intense ionospheric currents in the North American
sector auroral zone launched TIDs that were observed by the midlatitude SuperDARN radars and
the detrended TEC. Before 4:00 UT at radar frequency at 11.5 MHz, the Fort Hays West (FHW)
midlatitude radar beam 12 looking northwest over the central Canada observed the ionospheric
scatter showing enhancements in the positive LoS velocities (toward the radar; Fig. 9a) due to
fluctuating eastward ionospheric flows at the equatorward edge of an expanded dawn convection
cell associated with the fluctuating westward electrojet. The ionospheric currents were sensed by
magnetometers, including one in Fort Simpson (FSIM; www.carisma.ca/). The X component of
the ground magnetic field and time series of the latitudinal maxima in EICs at the longitude of
120°W, are superposed. After 14:00 UT, when the radar frequency was set to 15 MHz, the HF
propagation allowed to observe TIDs in the ground scatter. The ground scatter slant ranges
between 1000 and 3000 km correspond to the mapped ground scatter ranges between 200 and 1200
km.

Two major intensifications of the westward electrojet at ~13:10 and 14:10 UT (Fig. 9b) launched
LSTIDs observed in the ground scatter starting at ~14:00 and 15:00 UT.  The mapped EICs in Fig.
10a show the first major intensification of the westward electrojet (the EIC maxima at each
longitude are highlighted) that was followed by the second intensification an hour later (Fig. 9b).
They launched equatorward propagating LSTIDs (wavelength greater than 1000 km) observed in
the detrended vTEC maps (Fig. 10b). Fig. 11a shows the LSTIDs between 13:00 and 16:00 UT in
the detrended vTEC mapped along longitude of 100°W in the North American sector. The
approximate equatorward propagation speeds were between 500 and 800 m/s.

In the European sector, during the most disturbed time following the HSS/CIR arrival on
November 4 the dayside auroral oval expanded down to latitude ~63°N. Fig. 11b shows the
equivalent ionospheric currents estimated from the IMAGE magnetometers. LSTIDs (Fig. 10c)
that were launched by intensifications of the east electrojet were observed propagating
equatorward at speeds between 400 and 500 m/s in mid latitudes are mapped along longitude of
15°W between 11:00 and 18:00 UT (Fig. 11b).

In summary, the cases discussed in Sections 3.1 and 3.2 highlight the importance of solar wind
coupling to the M-I-T system, particularly on the dayside, in the generation of AGWs/TIDs. The
fluctuations of the IMF, sometimes Alfvénic, modulate PIFs and ionospheric currents in the cusp
launching GW/TIDs.

**4. AGWs/TIDs originating from sources in the troposphere**

MSTIDs caused by GWs with periods of 10-40 min propagating obliquely upward in the
thermosphere/ionosphere have been studied using multi-frequency and multi-point continuous HF
Doppler sounding system located in the western part of Czechia (Chum et al., 2021). The 2-D
propagation analysis of the HF Doppler sounders data is applied to available signals to exclude
data gaps and to select time intervals in which the phase shifts/time delays between signals
corresponding to different sounding paths (transmitter-receiver pairs) were approximately
constant.

The observed azimuths depend on season with southeastward propagation more likely in winter
months, suggesting that cold season low pressure systems in the northeast Atlantic are sources of
the GWs, which supports previously published results (Azeem et al., 2018) and points to winter
jet stream as a likely source of GWs. In this section we examine such cases and trace TIDs in the
detrended GNSS TEC maps (Fig. 12) and observed by the HF Doppler sounders propagating
eastward to southeastward from intensifying low-pressure weather systems over the east Atlantic.
The animated TEC maps can be viewed in the Supplement Video, which also shows the
equatorward propagating TIDs that originated at high latitudes as discussed in Section 3.

*4.1. Events of November 1-8, 2014*
Spectral and propagation analysis for all available 7.04 MHz signals from November 1 to 8, 2014
was performed (Fig. 13). Only daytime values are available because the critical frequency foF2 is
too low at night (most of the nights are also not available at 4.65 MHz). On November 6 an
enhanced noise (electromagnetic interference) prevented reliable analysis for a substantial part of
the day. Fig. 13b shows dynamic spectra (periodograms) of Doppler shift signal obtained as the
average of the maxima of three power spectral densities corresponding to three different
transmitter – receiver pairs (Section 2) shown in Fig. 13a (including artificial offsets). The
observed periods (Fig. 13b) range from 10 to about 40 min. The propagation azimuths (Fig. 13c)
were mostly from 100 to about 160° (waves propagating south-eastward). In all cases, the azimuth
is only plotted if the averaged Doppler fluctuations exceeded 0.12 Hz, the estimate of uncertainty
of azimuth is less than 10° and the estimate of uncertainty in velocity is less than 10%. The phase
velocities fluctuated typically between 100 and 200 m/s. Fig. 14 shows the analysis results on an
expanded time scale to better see the TID characteristics for November 8.

During the period from November 1 to 8, 2014, the south-eastward propagating MSTIDs were
observed by the HF sounders and detrended vTEC. Low-pressure systems deepening over the
North-east    Atlantic,    shown    in    the    surface    pressure    analysis    charts
([https://www.wetter3.de/archiv_ukmet_dt.html](https://www.wetter3.de/archiv_ukmet_dt.html)),    were    likely    sources    of    these    MSTIDs
propagating eastward to southeastward, as observed in the detrended vTEC maps (indicated by
arrows in Figs. 12a,b) on November 1 and 8, 2014. At the same time, the vTEC maps on both days
also reveal equatorward propagating TIDs at latitudes down to ~50°N that originated in the cusp
ionospheric footprint over Svalbard, as already discussed in Section 3.2.

The Doppler shift frequencies (Fig. 14a) recorded at frequency 7.04 MHz on November 1 and 8,
2014 show temporal evolution of power spectral densities (color-coded arbitrary units) of received
signals that correspond to three different transmitter-receiver pairs. There was enhanced noise due
to the electromagnetic interference on 8 November from about 9:30 to 12:30 UT. The straight
horizontal line in the upper signal trace in the spectrogram corresponds to ground wave from one
of the transmitters, located only ~7 km from the receiver. The middle and bottom signal traces in
the spectrogram correspond to other two transmitters. As described in more detail in Section 2.2,
the use of well correlated signals at two or three different frequencies makes it possible to
determine a 3-D phase velocity vector. The results that are summarized in Table S1 in the
Supplement separately for the observation at frequencies of 4.65 and 7.04 MHz show mostly
similar values of horizontal velocities (ranging from ~100 to 200 m/s) and azimuths (ranging from
~90 to 145°).

*4.2. Events of December 5-7, 2024*
Similarly to events discussed in Section 4.1, low-pressure systems deepening over the North-east
Atlantic on December 5-7 were likely sources of GWs driving the observed southeastward
propagating MSTIDs (Fig. 12c). To further support the results in Section 4.1 we analyzed these
recent events recorded at 4.65 and 7.04 MHz. Due to the critical frequency of the ionosphere 4.65
MHz allows to obtain longer time intervals with usable signals but with smaller amplitudes. The
signals at 7.04 MHz reflect from higher altitudes where the amplitudes are larger. The results of
the spectral and propagation analysis for all available 7.03 MHz signals for December 5-7, 2024
are shown in Fig. 15. Fig. 15b shows dynamic spectra of Doppler shift signal obtained as the
average of the frequencies corresponding to the maxima of the three power spectral densities for
the three different transmitter – receiver pairs shown in Fig. 15a (including artificial offsets, which
were removed before the spectral and propagation analysis). The observed periods of TIDs/GWs
(Fig. 15b) range from a few minutes up to an hour. The propagation azimuths (Fig. 15c) were
mostly around 140 deg. (southeastward) and phase velocities fluctuated typically between 150 and
290 m/s.

*4.3. Comparison between TIDs observed in the detrended vTEC maps and by the HF Doppler*
*sounding system*
The Doppler shift spectrograms recorded at frequency 7.04 MHz on November 8, 2014 and
December 5, 2024 are shown in Fig. 16a (top panels). In Fig. 16b (middle panels), the detrended
vTEC mapped along the latitude of 50° shows eastward propagating TIDs towards the longitude
of the HF sounding system with estimated approximate velocities shown. The bottom panels (Fig.
16c) show time series of the detrended vTEC at longitude of 7°E with the normalized FFT spectra
shown in the insets.

In November 2014, the amplitudes of MSTIDs observed in the detrended vTEC and by the HF
Doppler sounder were relatively small, and the coverage by GNSS receivers at longitudes greater
than 10°E was sparse. In contrast, in December 2024, the GNSS coverage significantly improved,
and the amplitudes of MSTIDs observed both in the vTEC and by HF Doppler sounder were much
larger.

While it is difficult to compare in detail the results obtained by these two very different techniques
(one based on the global mapping of vTEC using the thin shell approximation at 350-km altitude,
and the other local measurements dependent on the HF propagation and the critical frequency for
reflection altitude), the phase velocities (Fig. 16b) and periods (Fig. 16c) of TIDs estimated from
the detrended vTEC are similar to those obtained by the HF Doppler sounder as discussed in
Sections 4.1 and 4.2 (Figs, 14 and 15).

More cases of eastward to south-eastward propagating MSTIDs observed by the HF Doppler
sounder and in the detrended vTEC observed in 2014 on November 1, 3 and 7 (see Fig. S6-S8 in
the Supplement), November 22 and 24, December 9-10, and December 24, were also associated
with intense low-pressure systems in the North-east Atlantic.

*4.4. Physical mechanism of GW generation in the troposphere*
While tropospheric convection is a common source of gravity waves (e.g., Azeem, 2021), no deep
convection could be identified in the cold fronts of low-pressure systems over the North-east
Atlantic (GIBBS). Mesoscale gravity waves generated by geostrophic adjustment processes and
shear instability have been observed (Uccellini and Koch, 1987; Koch and Dorian, 1988).
Plougonven and Zhang (2014) reviewed the current knowledge and understanding of gravity
waves near jets and fronts. Plougonven and Teitelbaum (2003; their Figure 2) showed patterns of
alternating bands of convergence and divergence in maps of divergence of the horizontal wind for
the lower stratosphere, which have been interpreted as the signature of inertia-gravity waves
propagating upwards above the tropopause. A conceptual model of a common synoptic pattern has
been identified with a source of gravity waves near the axis of inflection in the 300-hPa
geopotential height field (Koch and O'Handley, 1997; their Figure 2).

In Sections 4.1 and 4.2, the cases of MSTIDs on November 1 and 8, 2014 and December 5, 2024
propagating eastward to southeastward observed in the detrended vTEC maps (Fig. 12) and by the
HF Doppler sounding system (Figs. 13, 14 and 15) are attributed to sources in the troposphere,
namely deepening low pressure weather systems. This is consistent with the conceptual model
referenced above. Using the ERA5 reanalysis (Hersbach *et al.*, 2020), Fig. 17 shows the 300-hPa
geopotential height, approximate axis of inflection (a probable source region of gravity waves that
is indicated by red dashed line), and horizontal winds at 300 hPa for the three events in 2014 and
2024. Fig. 17b shows the divergence of the horizontal wind at 150-hPa level. The alternating bands
of convergence and divergence are similar to those interpreted by Plougonven and Teitelbaum
(2003) as gravity waves propagating to the lower stratosphere. Other cases of MSTIDs on
November 3 and 7 can be viewed in Figs. S6 and S7 in the Supplement. Mesoscale GWs
propagating eastward and upward into the stratosphere generated by geostrophic adjustment
processes and shear instability may be common and could be driving MSTIDs.

**5. Discussion**
The solar wind – MIT coupling is known to modulate the intensity of ionospheric currents,
including the auroral electrojets, which in turn launch atmospheric gravity waves causing TIDs.
The cases of dayside equatorward propagating TIDs were observed with PFISR, SuperDARN, and
detected in the detrended GNSS vTEC maps. This is consistent with previously published results
and interpretations (e.g., Prikryl et al. 2022; and references therein). The dayside TIDs are
commonly generated in the ionospheric footprint of the cusp. They were observed every day over
Alaska during the PFISR experiment (8-15 January 2013) and in Europe (1-8 November 2014).

In Section 3.1, we have shown evidence that even during a geomagnetically very quiet period the
TIDs that were observed by PFISR in Alaska can be attributed to sources at high latitudes (Fig. 3).
Quasiperiodic intensifications of the high-latitude ionospheric convection that were the source of
these TIDs were observed poleward of Alaska over the East Siberian and Beaufort Seas. The
ionospheric currents associated with PIFs could not be detected by ground magnetometers, and the
*Kp* index indicated a quiet period. The ionospheric footprint of the cusp where the pulsed
ionospheric flows and associated currents are sources of TIDs may be located further poleward of
any ground magnetometers. Low geomagnetic activity is often taken as a pretext to dismiss auroral
sources of TIDs. Throughout the PFISR experiment from January 8 to 15, 2013 the TIDs that were
observed on the dayside resulted from the solar wind-MIT coupling that modulated PIFs in the
ionospheric cusp footprint poleward of Alaska. Similarly, in the European sector.
The importance of solar wind coupling to the M-I-T system, particularly on the dayside, for the
generation of AGWs/TIDs (Prikryl et al. 2005) is further highlighted in Section 3.2 by cases where
solar wind Alfvén waves modulated PIFs and ionospheric currents in the cusp launching
AGWs/TIDs.
Regarding TIDs originating from the troposphere, there has been plentiful evidence of neutral
atmosphere-ionosphere coupling via atmospheric gravity waves propagating into the upper
atmosphere from sources in the lower atmosphere including convective storms (Alexander, 1996).
Azeem and Barlage (2018) and Vadas and Azeem (2021) presented cases of convective storm
generating TIDs, which exhibited partial to full concentric, or almost plane-parallel phase fronts.
The latter TIDs were generated by extended squall line (Azeem and Barlage, 2018). However, in
the cases discussed in Section 4.1 there was no significant convection in the cold fronts that would
generate such TIDs. The eastward propagating MSTIDs observed in the detrended vTEC maps
and by the HF originated from low pressure sounding system were likely driven by GWs generated
by geostrophic adjustment processes and shear instability in the troposphere. Geostrophic
adjustment processes (Uccellini and Koch, 1987) generating gravity waves have not been
previously considered as sources of TIDs. This is supported by the ERA5 meteorological
reanalysis showing the GWs in the lower stratosphere.

The occurrence of eastward propagating MSTIDs at mid latitudes over Europe is very common in
the cold season, and their association with low-pressure systems can be readily seen browsing the
archive of detrended TEC maps (https://aer-nc-web.nict.go.jp/GPS/EUROPE/MAP/#2014). An
assortment of such cases for December 2014 can be viewed in the Supplement Figures S9-S13. At
the same time, the animations of TEC maps provided in the archive show equatorward propagating
TIDs, including LSTIDs, originating at high latitudes that are generated by solar wind-M-I-T
coupling. Dayside equatorward propagating TIDs that were observed in the detrended vTEC in
Europe (15°E), the U.S. (100°W) and Alaska (148°W) on December 5, 2024 can be viewed in the
Supplement Figure S14.

In this study, a multi-instrument approach is used to trace/attribute the observed TIDs to sources
of AGWs in the upper and lower atmosphere, and to identify physical mechanisms. The solar wind
coupling to the M-I-T system generates equatorward propagating TIDs on the dayside even during
geomagnetically quiet conditions. This is important, because the auroral sources of TIDs observed
during quiet conditions are often not considered or dismissed. On the other hand, intensifying low
pressure weather systems generate AGWs propagating to the lower stratosphere and beyond,
driving TIDs even when there is no significant tropospheric convection. More work needs to be
done to better understand such cases, and many aspects of the system as a whole should be
considered when determining the source of TIDs, as simple metrics/indices hide critical details.

**6. Summary and conclusions**
Traveling ionospheric disturbances are observed by radars, Doppler sounders, and the GNSS TEC
mapping technique. Medium- to large-scale TIDs propagating equatorward were generated by
solar wind coupling to the dayside magnetosphere-ionosphere-thermosphere modulating
ionospheric convection and currents, including auroral electrojets. MSTIDs that were observed
over Alaska by the Poker Flat incoherent scatter radar and by two SuperDARN radars are attributed
to gravity waves generated in the ionospheric cusp footprint poleward of Alaska even when
geomagnetic activity was low. Dayside MSTIDs were observed every day over Alaska during the
7-day PFISR experiment. In another case, following the arrival of high-speed solar wind stream
that triggered a minor geomagnetic storm, major intensifications of the westward electrojet over
the North American sector launched LSTIDs observed by a mid-latitude SuperDARN radar and in
the detrended global TEC maps. In the European sector, the equatorward propagating TIDs are
attributed to solar wind Alfvén waves coupling to the dayside magnetosphere modulating
ionospheric convection and currents in the cusp footprint over Svalbard. On the other hand, the
cases of eastward to southeastward propagating MSTIDs observed at mid latitudes in the detrended
GNSS TEC maps and by the HF Doppler sounders in Czechia originated from low pressure
weather systems intensifying in the north-east Atlantic. Gravity waves propagating from the
troposphere and lower stratosphere were likely generated by geostrophic adjustment processes,
which have not been linked to TIDs previously.

*Data availability.* The solar wind data are provided by the NSSDC OMNI
(http://omniweb.gsfc.nasa.gov; NASA, 2022). The ground-based magnetometer data are
archived at the website of the Canadian Array for Realtime Investigations of Magnetic Activity
(CARISMA) (https://www.carisma.ca/; University of Alberta, 2022), and the IMAGE website at
https://space.fmi.fi/image/www/index.php?. The PFISR data are available
at https://data.amisr.com/database/61/cal/2014/11/. SuperDARN data are available
at https://www.frdr-dfdr.ca/repo/collection/superdarn (FRDR, 2022). Line-of-Sight TEC data can
be acquired from the Madrigal database (http://cedar.openmadrigal.org/; CEDAR, 2022) and
CHAIN GNSS data are available at http://chain.physics.unb.ca/chain/pages/data_download
(CHAIN, 2022).
Equivalent Ionospheric Currents (EICs) derived by the Spherical Elementary Currents Systems
(SECS) technique are archived at http://vmo.igpp.ucla.edu/data1/SECS/ (SECS, 2022)
and https://cdaweb.gsfc.nasa.gov/pub/data/aaa_special-purpose-datasets/spherical-elementary-
and-equivalent-ionospheric-currents-weygand/; https://doi.org/10.21978/P8D62B, Weygand,
2009a; https://doi.org/10.21978/P8PP8X, Weygand, 2009b). The Czech HF Doppler shift
spectrograms can be found in the archive at http://datacenter.ufa.cas.cz/.
GNSS data for this study were provided by the following organizations: International GNSS
Service (IGS), UNAVCO (https://www.unavco.org/data/gps-gnss/gps-gnss.html), Dutch
Permanent GNSS Array (http://gnss1.tudelft.nl/dpga/rinex), Can-Net (https://www.can-net.ca/),
Scripps Orbit and Permanent Array Center (Garner, http://garner.ucsd.edu/pub/), French Institut
Geographique National, Geodetic Data Archiving Facility (GeoDAF,
http://geodaf.mt.asi.it/index.html), Crustal Dynamics Data Information System (CDDIS,
https://cddis.nasa.gov/archive/gnss/data/daily/), National Geodetic Survey
(https://geodesy.noaa.gov/corsdata/), Instituto Brasileiro de Geografia e Estatistica
(http://geoftp.ibge.gov.br/informacoes_sobre_posicionamento_geodesico/rbmc/dados/), Instituto
Tecnologico Agrario de Castilla y Leon (ITACyL, ftp://ftp.itacyl.es/RINEX/), TrigNet South
Africa (ftp://ftp.trignet.co.za), The Western Canada Deformation Array (WCDA,
ftp://wcda.pgc.nrcan.gc.ca/pub/gpsdata/rinex), Canadian High Arctic Ionospheric Network
(CHAIN, http://chain.physics.unb.ca/chain/pages/data_download), Pacific Northwest Geodetic
Array (PANGA, http://www.geodesy.cwu.edu/pub/data/), Centro di Ricerche Sismologiche,
Système d'Observation du Niveau des Eaux Littorales (SONEL, ftp://ftp.sonel.org/gps/data),
INGV - Rete Integrata Nazionale GPS (RING, http://ring.gm.ingv.it/),RENAG : REseau
NAtional GPS permanent (http://rgp.ign.fr/DONNEES/diffusion/), Australian Space Weather
Services (https://downloads.sws.bom.gov.au/wdc/gnss/data/),GeoNet New Zealand
(https://www.geonet.org.nz/data/types/geodetic), National Land Survey Finland (NLS,
https://www.maanmittauslaitos.fi/en/maps-and-spatial-data/positioning-services/rinex-palvelu),
SWEPOS Sweden (https://swepos.lantmateriet.se/), Norwegian Mapping Authority (Kartverket,
https://ftp.statkart.no/), Geoscience Australia (http://www.ga.gov.au/scientific-
topics/positioning-navigation/geodesy/gnss-networks/data-and-site-logs), Institute of
Geodynamics, National Observatory of Athens (https://www.gein.noa.gr/services/GPSData/),and
European Permanent GNSS Network (EUREF,
https://www.epncb.oma.be/_networkdata/data_access/dailyandhourly/datacentres.php).
*Author contributions*. PP and RGG contributed to conception and design of the study. PP, DRT,
JC, SC, RGG, and JMW acquired the resources and contributed to methodology, software, specific
data analysis, visualization, and organized the databases. PP wrote the first draft of the manuscript.
All authors contributed to manuscript revision and approved the submitted version.
*Competing interests*. The authors declare that they have no conflict of interest.
*Acknowledgments*. Infrastructure funding for CHAIN was provided by the Canada Foundation for
Innovation and the New Brunswick Innovation Foundation. CHAIN operation is conducted in
collaboration with the Canadian Space Agency (CSA). We are grateful to the Australian Bureau
of Meteorology, Space Weather Services for the provision of GNSS data. CDDIS is one of the
Earth Observing System Data and Information System (EOSDIS) Distributed Active Archive
Centers (DAACs), part of the NASA Earth Science Data and Information System (ESDIS) project.
Datasets and related data products and services are provided by CDDIS, managed by the NASA
ESDIS project. This material is based on services provided by the GAGE Facility, operated by
UNAVCO, Inc., with support from the National Science Foundation and the National Aeronautics
and Space Administration under NSF Cooperative Agreement EAR-1724794. A. Contributions by
the ACE (Norman F. Nees at Bartol Research Institute, David J. McComas at SWRI), NASA's
SPDF/CDAWeb, and the NSSDC OMNIWeb are acknowledged. The PFISR was developed under
NSF cooperative agreement ATM-0121483, and the data collection and analysis were supported
under NSF cooperative agreement ATM-0608577. The authors acknowledge the use of
SuperDARN data. SuperDARN is a collection of radars funded by the national scientific funding
agencies of Australia, Canada, China, France, Italy, Japan, Norway, South Africa, United
Kingdom, and the United States of America. The Fort Hays SuperDARN radars are maintained
and operated by Virginia Tech under support by NSF grant AGS-1935110. The King Salmon and
Prince George radars are operated under support of NSF grant AGS--2125323 from the Upper
Atmospheric Facilities Program, and by the Canada Foundation for Innovation, Innovation
Saskatchewan and the Canadian Space Agency, respectively. We thank the many different groups
operating magnetometer arrays for providing data for this study, including the THEMIS UCLA
magnetometer network (Ground-based Imager and Magnetometer Network for Auroral Studies).
The AUTUMNX magnetometer network is funded through the Canadian Space Agency/Geospace
Observatory (GO) Canada program, Athabasca University, Centre for Science/Faculty of Science
and Technology. The Magnetometer Array for Cusp and Cleft Studies (MACCS) array is
supported by the US National Science Foundation grant ATM-0827903 to Augsburg College. The
Solar and Terrestrial Physics (STEP) magnetometer file storage is at the Department of Earth and
Planetary Physics, University of Tokyo and maintained by Kanji Hayashi (hayashi@grl.s.u-
tokyo.ac.jp). The McMAC Project is sponsored by the Magnetospheric Physics Program of
National Science Foundation through grant AGS-0245139. The ground magnetic stations are
operated by the Technical University of Denmark, National Space Institute (DTU Space). The
IMAGE magnetometer stations are maintained by 10 institutes from Finland, Germany, Norway,
Poland, Russia, Sweden, Denmark, and Iceland. The Canadian Space Science Data Portal is funded
in part by the Canadian Space Agency contract numbers 9 F007-071429 and 9 F007-070993. The
Canadian Magnetic Observatory Network (CANMON) is maintained and operated by the
Geological Survey of Canada. David R. Themens's contribution to this work is supported in part
through CSA grant no. 21SUSTCHAI and through the United Kingdom Natural Environment
Research Council (NERC) EISCAT3D: Fine-scale structuring, scintillation, and electrodynamics
(FINESSE) (NE/W003147/1) and DRivers and Impacts of Ionospheric Variability with EISCAT-
3D (DRIIVE) (NE/W003368/1) projects. James M. Weygand acknowledges NASA grant:
80NSSC18K0570, 80NSSC18K1220, NASA contract: 80GSFC17C0018 (HPDE), NAS5-
02099(THEMIS). Shibaji Chakraborty thanks the National Science Foundation for support under
grant AGS-1935110.

*Financial support.* David R. Themens's contribution to this work is supported in part through CSA grant no. 21SUSTCHAI and through the United Kingdom Natural Environment Research Council (NERC) EISCAT3D: Fine-scale structuring, scintillation, and electrodynamics (FINESSE) (NE/W003147/1) and DRivers and Impacts of Ionospheric Variability with EISCAT-3D (DRIIVE) (NE/W003368/1) projects. J. Chum was funded by T-FORS project by European Commission (number SEP 210818055) and by the Johannes Amos Comenius Programme (P JAC), project No. CZ.02.01.01/00/22_008/0004605, Natural and anthropogenic georisks". James M. Weygand is supported by the NASA grant: 80NSSC18K0570, 80NSSC18K1220, NASA contract: 80GSFC17C0018 (HPDE), NAS5-02099(THEMIS). Shibaji Chakraborty is supported by the National Science Foundation under grant AGS-1935110.

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

**Figures**

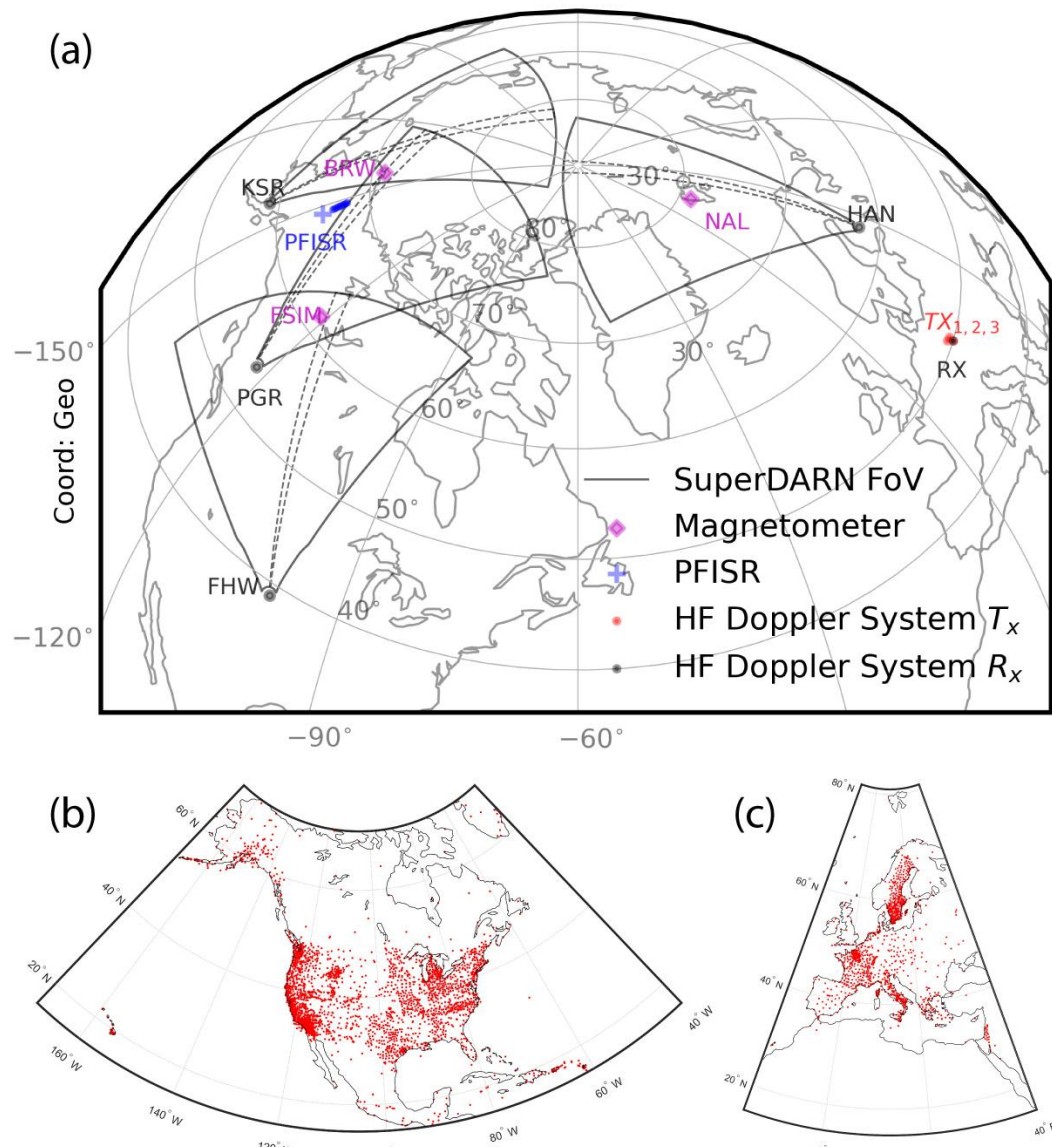

**Figure 1: (a)** The field-of-views of SuperDARN radars in King Salmon (KSR), Prince George
(PGR) Fort Hayes West (FHW) and Hankasalmi (HAN) are shown in black color. The Poker
Flats Incoherent Scatter Radar (PFISR) beam 2 range gates footprints are shown in blue color.
Also, the ground magnetometers in Barrow (BRW), Fort Simpson (FSIM) and Ny Ålesund
(NAL) and locations of transmitters ($T_X$) and a receiver ($R_X$) of the HF Doppler sounding system
operating in the Czech Republic are shown. **(b, c)** Examples of the GNSS station distribution in
the two local domains of interest, when there were 5708 stations available globally.


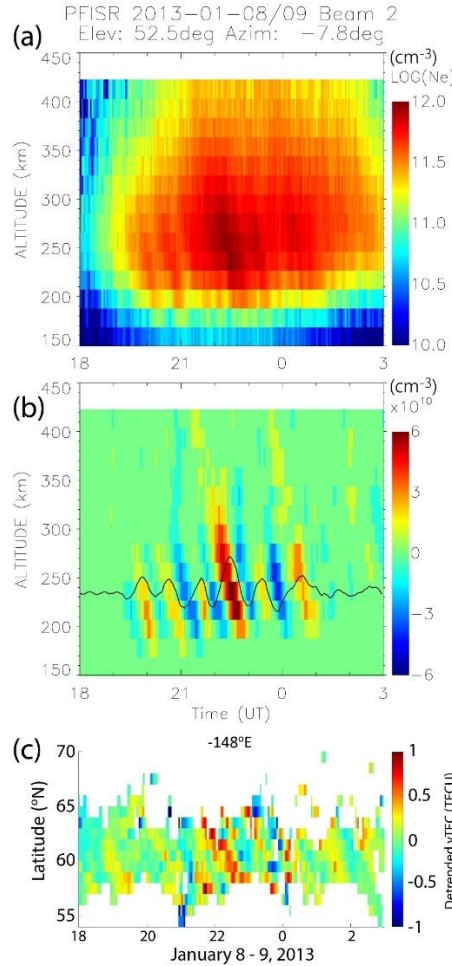

**Figure 2: (a)** Ionospheric density observed by the PFISR radar beam 2 and **(b)** detrended using a
33-point wide Savitzky-Golay filter. **(c)** The detrended GNSS vTEC mapped at latitude bins
along the longitude of PFISR.

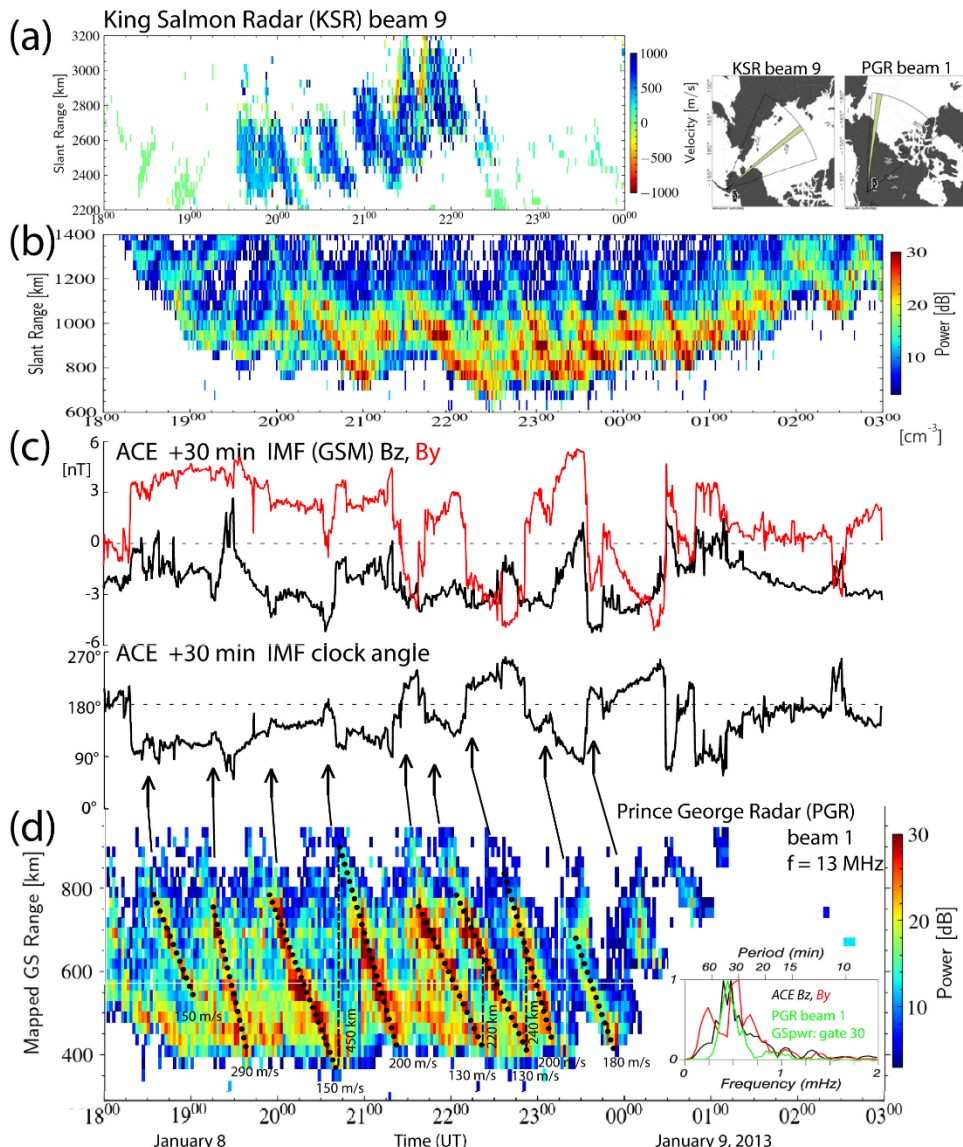

**Figure 3: (a)** The line-of-sight velocities and **(b)** the sea scatter power as a function of the slant
range observed by the KSR radar beam 9. **(c)** The time-shifted time series of the IMF $B_z$, $B_y$, and
the IMF clock angle observed by ACE spacecraft. **(d)** The mapped ground scatter power observed
by the PGR radar beam 1. The arrows indicate southward turning of the IMF. Normalized FFT
spectra of the detrended IMF $B_z$, $B_y$, and the Prince George radar ground scatter power (beam 1,
gate 30, mapped GS range 730 km) are shown in the inset. The estimated TIDs wavelengths and
phase velocities are shown.


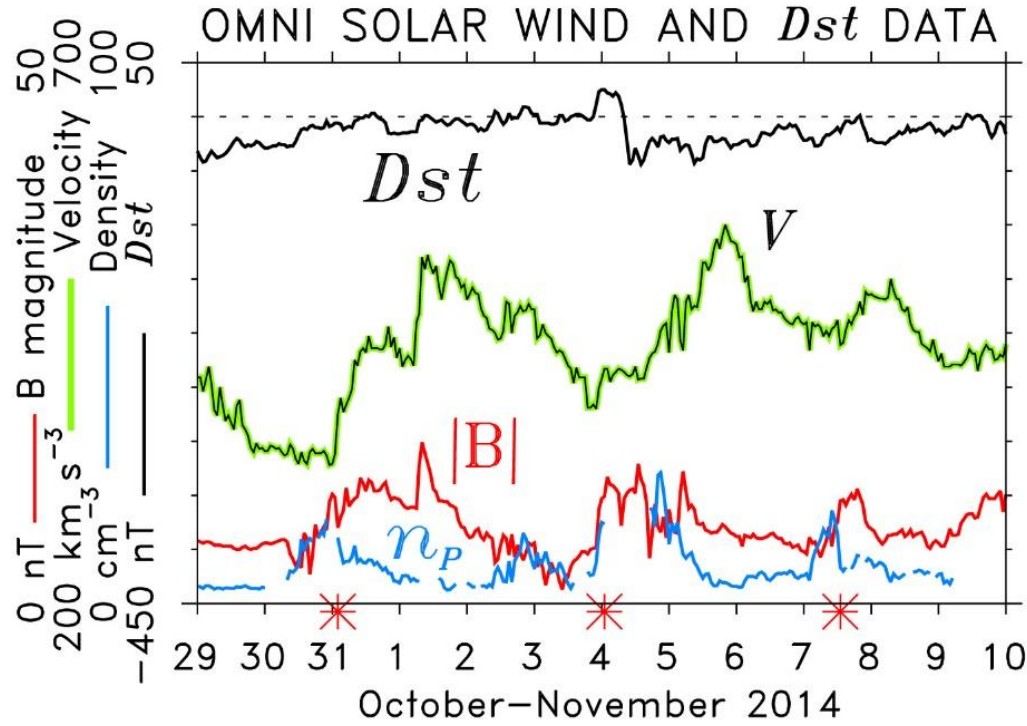

**Figure 4:** The OMNI solar wind velocity *V*, magnetic field magnitude |B|, and proton density $n_p$
showing three HSS/CIRs on October 31, November 4 and 7 are marked by red asterisks at the
time axis. The ring current *Dst* index is also shown.

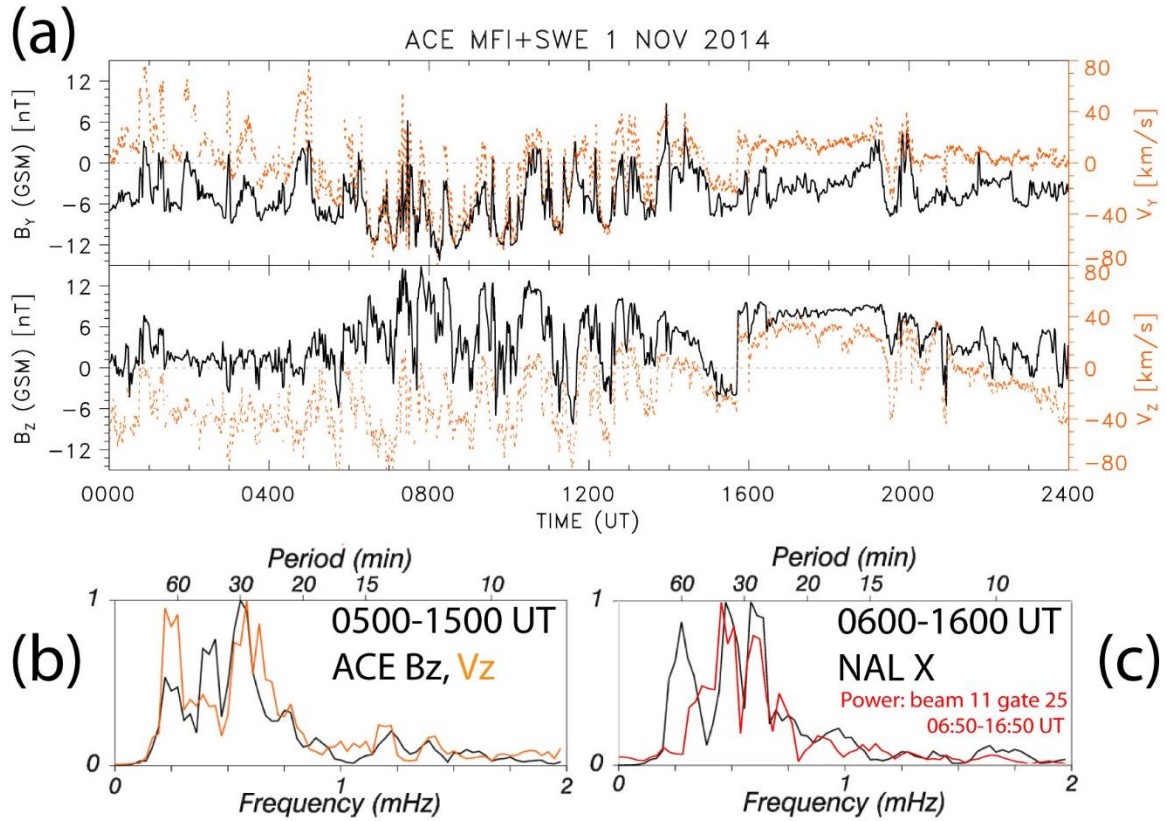

**Figure 5: (a)** The components of the magnetic field and solar wind velocity observed by ACE,
**(b)** the FFT spectra of the detrended time series of IMF $B_z$ and solar velocity $V_z$, and **(c)** the FFT
spectra of the time series of the X-component of ground magnetic field perturbations in Ny
Ålesund (NAL) and the Hankasalmi radar ground scatter power (beam 11, gate 25, slant range
1305 km).

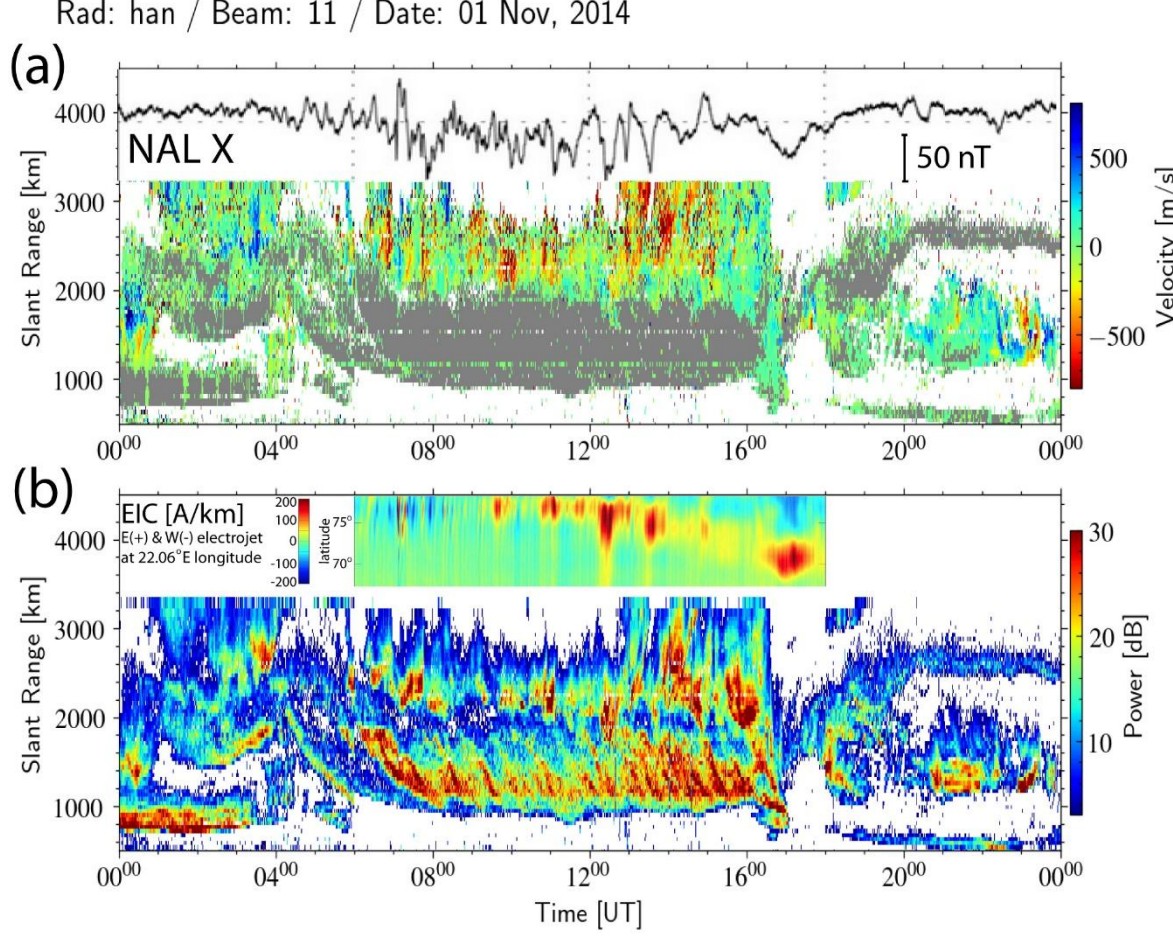

Rad: han / Beam: 11 / Date: 01 Nov, 2014

**Figure 6: (a)** The line-of-sight (LoS) velocity and **(b)** the radar scatter power (ground scatter
power shown in grey color in the velocity plot) observed by the Hankasalmi radar beam 11 on
November 1, 2014. The X-component of the ground magnetic field perturbations in Ny Ålesund
(NAL) is superposed representing the fluctuations of ionospheric currents modulated by solar
wind Alfvén waves. 1D equivalent currents estimates that use all IMAGE magnetometers are
also superposed.

Rad: han / Beam: 11 / Date: 05 Nov, 2014

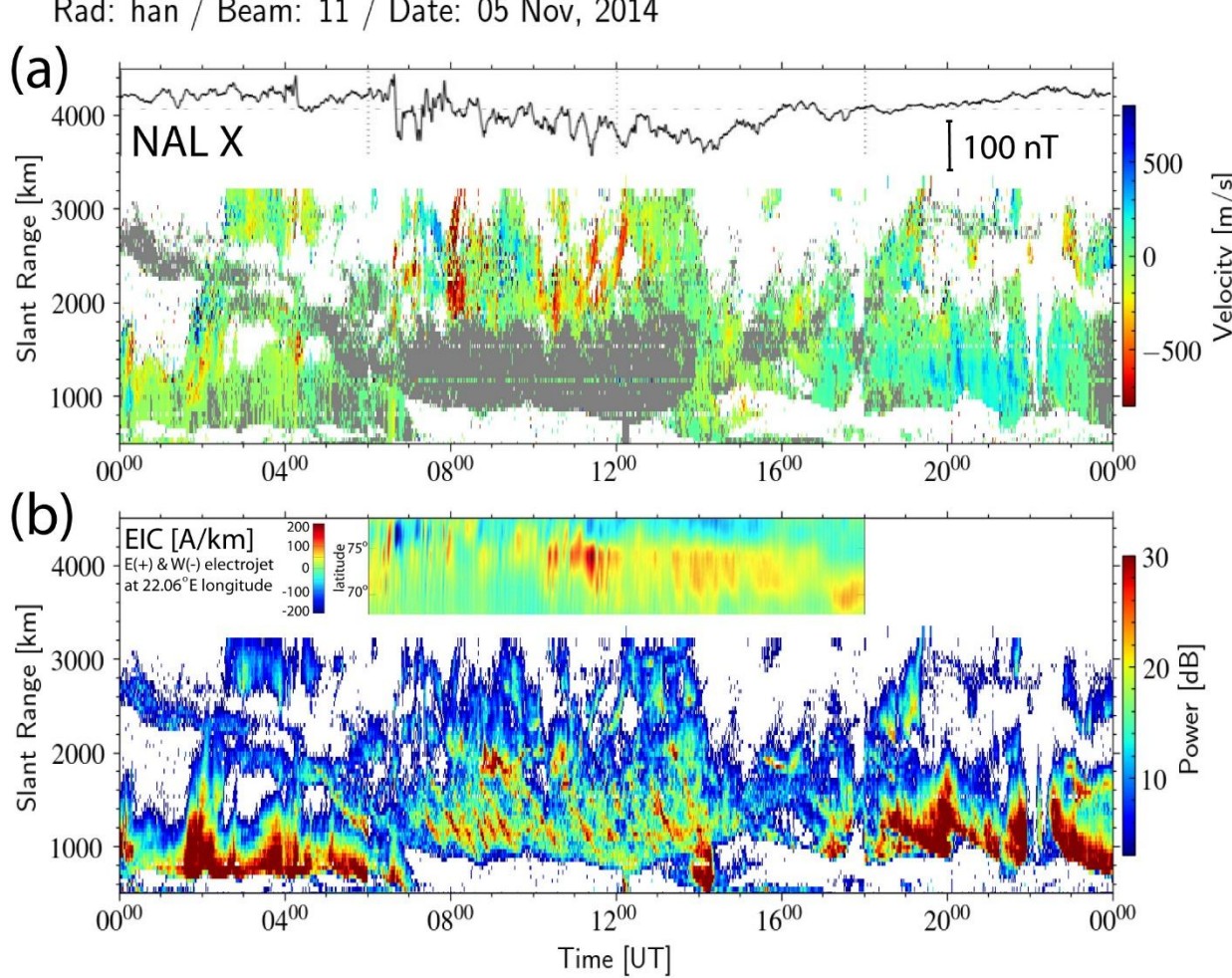

**Figure 7:** The same as Fig. 6 but for November 5, 2014.

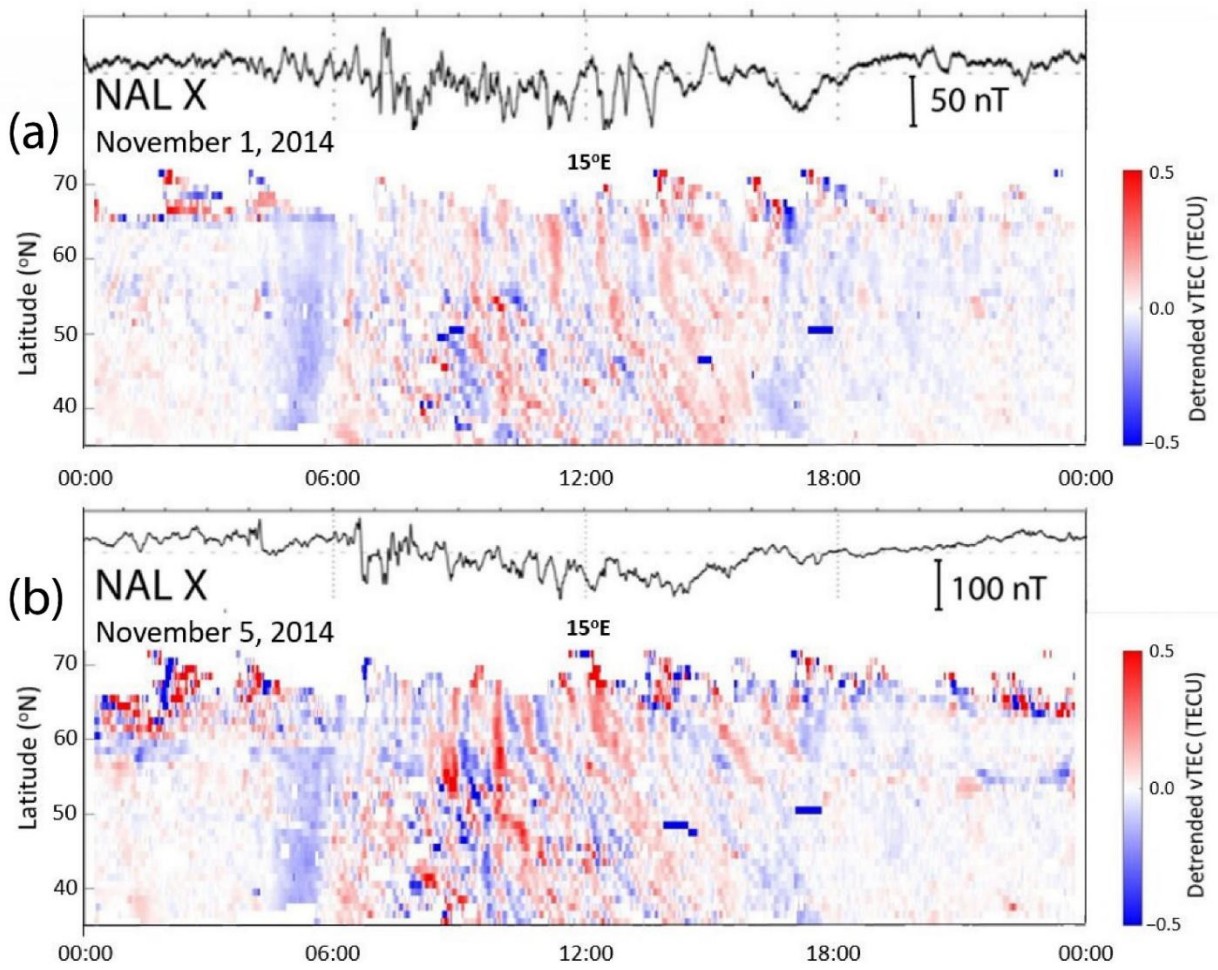

**Figure 8:** The detrended vTEC mapped along longitude of 15°E on **(a)** November 1 and **(b)**
November 5, 2014. The X-component of the ground magnetic field perturbations in Ny Ålesund
(NAL) is superposed.

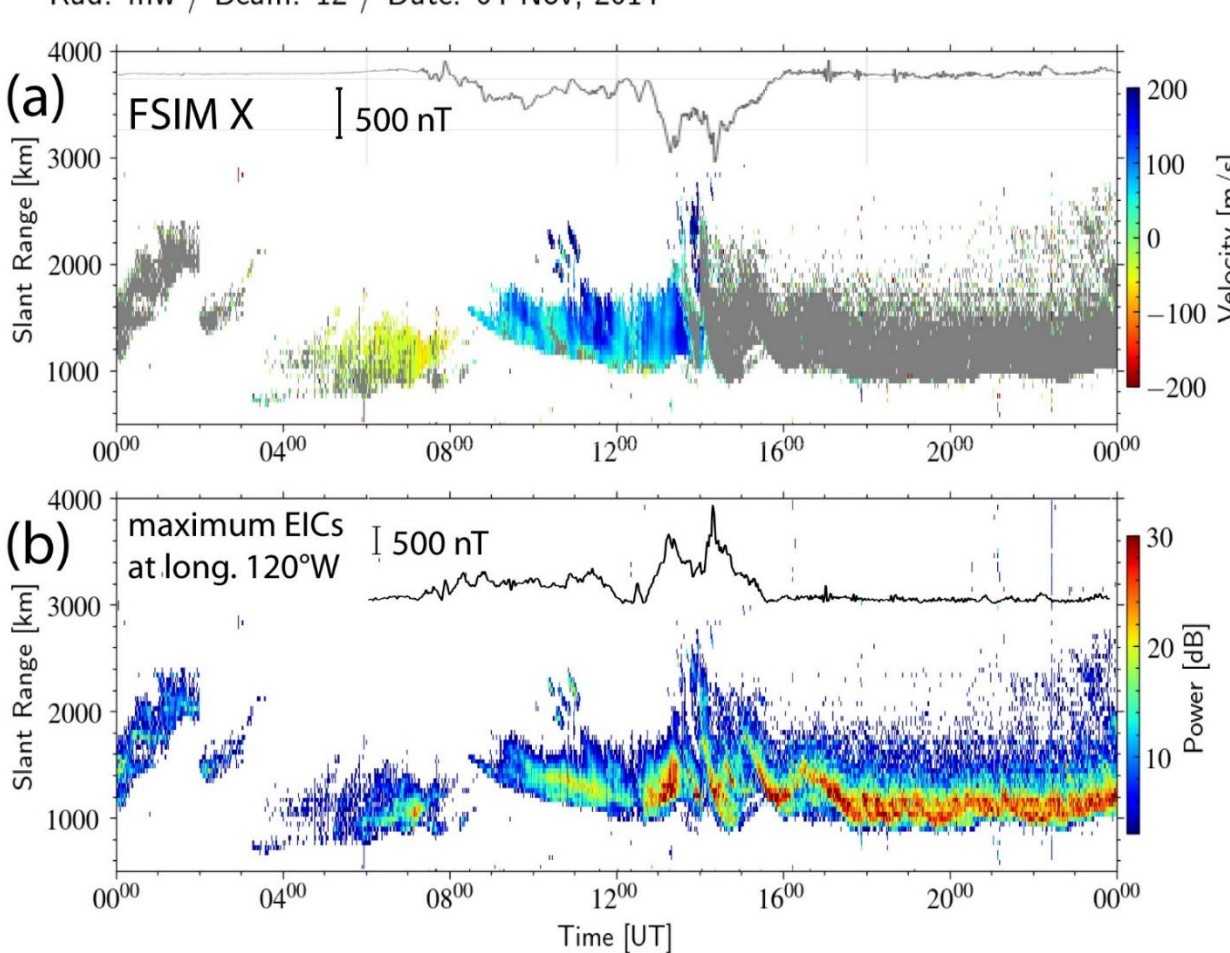

**Figure 9: (a)** The line-of-sight (LoS) velocity and **(b)** the radar scatter power (ground scatter
power shown in grey color in the velocity plot) observed by the Fort Hays West radar beam 12
on November 4, 2014. The X-component of the ground magnetic field perturbations in Fort
Simpson (FSIM) and the maximum EICs at longitude 120°W are superposed.

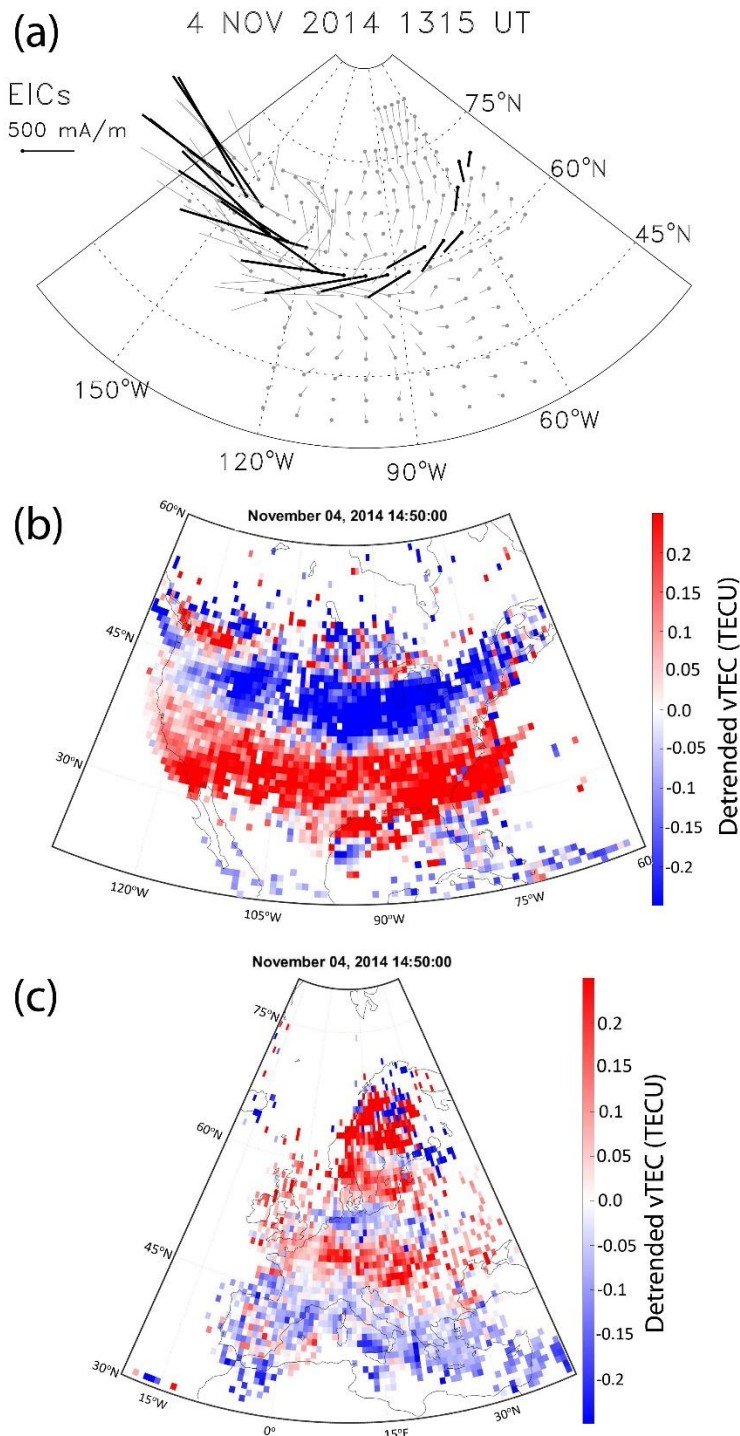

**Figure 10: (a)** The intensification of the westward electrojet over North America, and LSTIDs in
the detrended vTEC maps over **(b)** North America and **(c)** Europe.

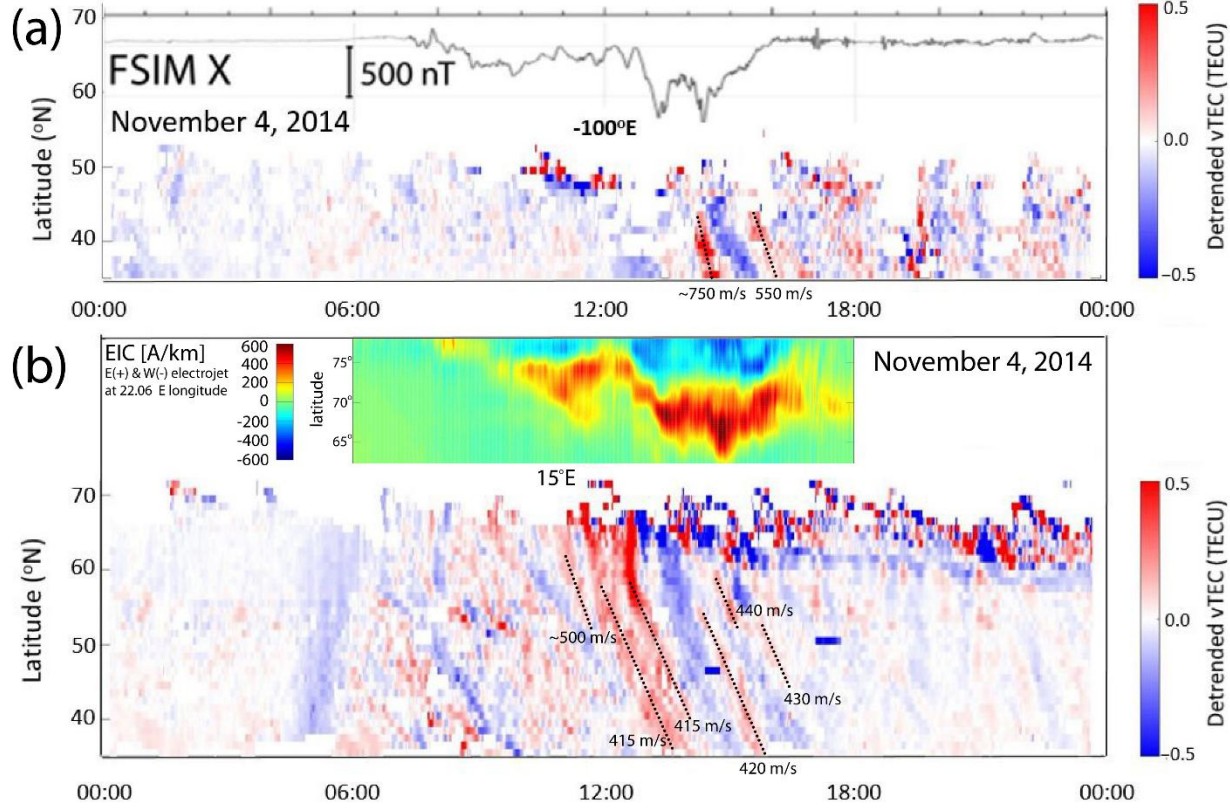

**Figure 11:** The detrended vTEC mapped along longitude of **(a)** 100°W and **(b)** 15°E on
November 4, 2014. The X-component of the ground magnetic field in Fort Simpson (FSIM) and
1D equivalent currents estimates over Scandinavia are superposed.

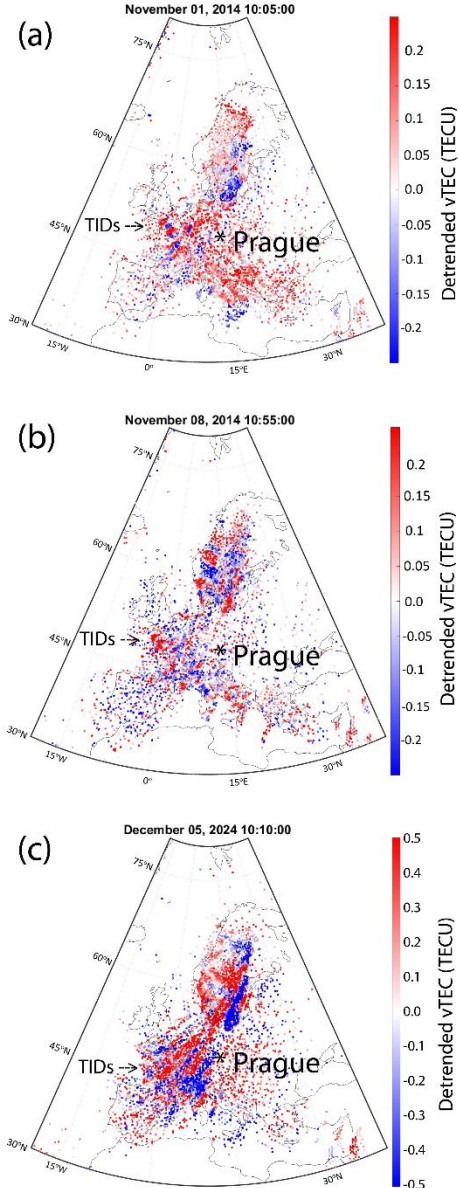



**Figure 12:** The detrended vTEC maps on **(a)** November 1, **(b)** November 8, 2014, and **(c)**
December 5, 2024.

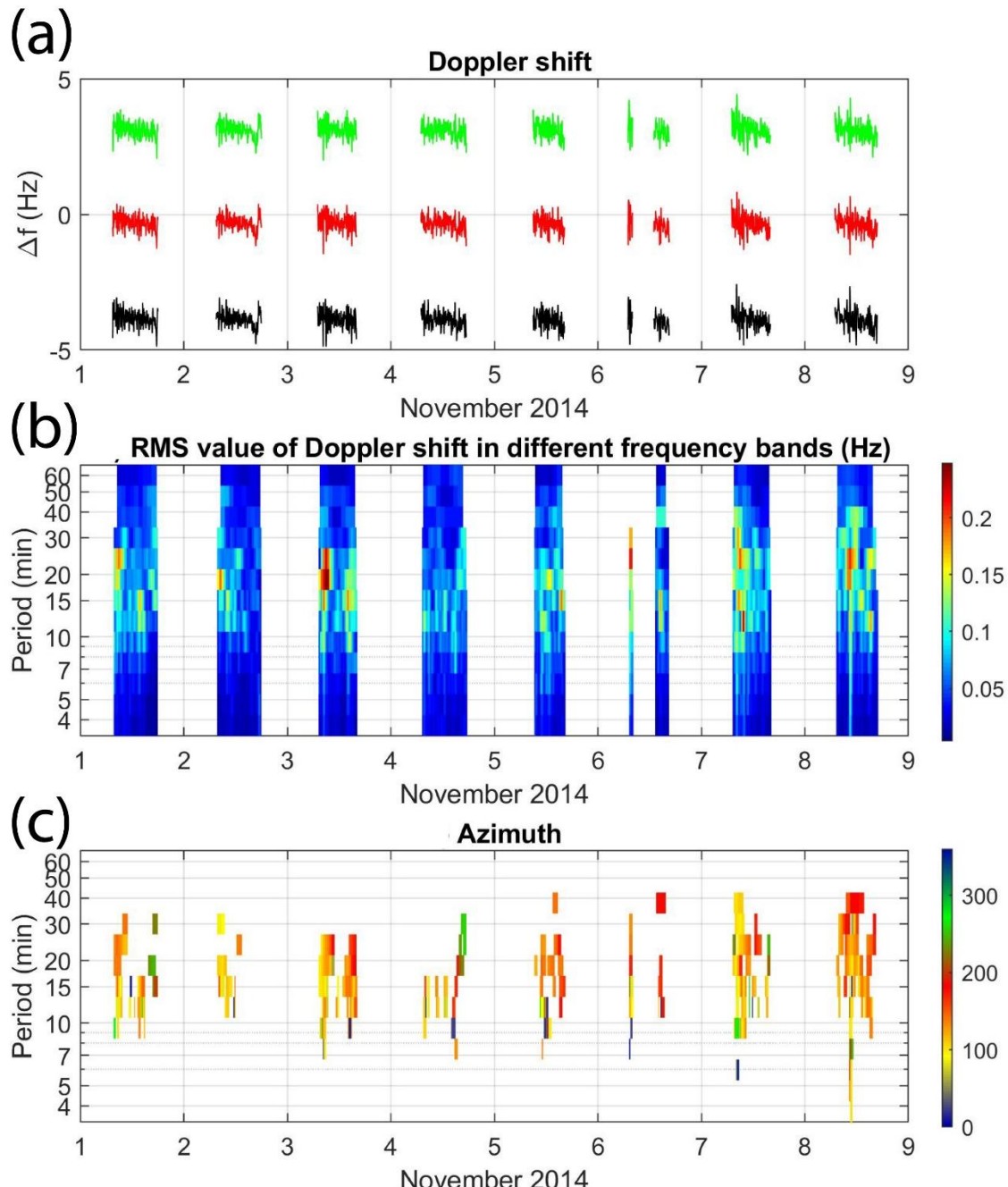

**Figure 13: (a)** Doppler shift frequencies of spectral density maxima for individual transmitter-
receiver pairs (including artificial offsets) from X to Y. **(b)** Dynamic spectra (periodograms) of
Doppler shift signals and **(c)** the propagation azimuth of waves, displayed as function of period
and time for November 1-8, 2014.

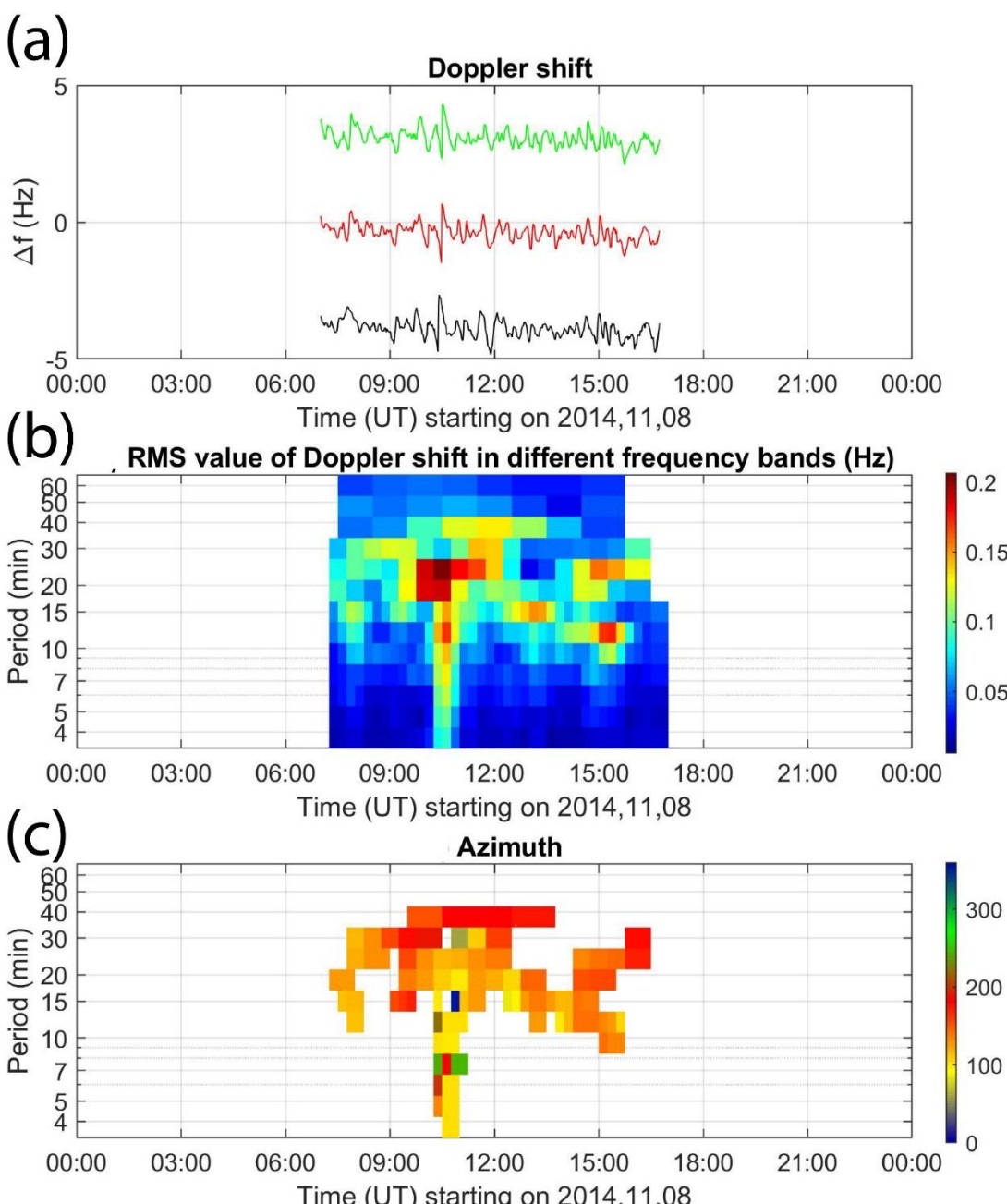

1175
1176    **Figure 14:** The same as Fig. 12 but expanded for November 8, 2014.
1177

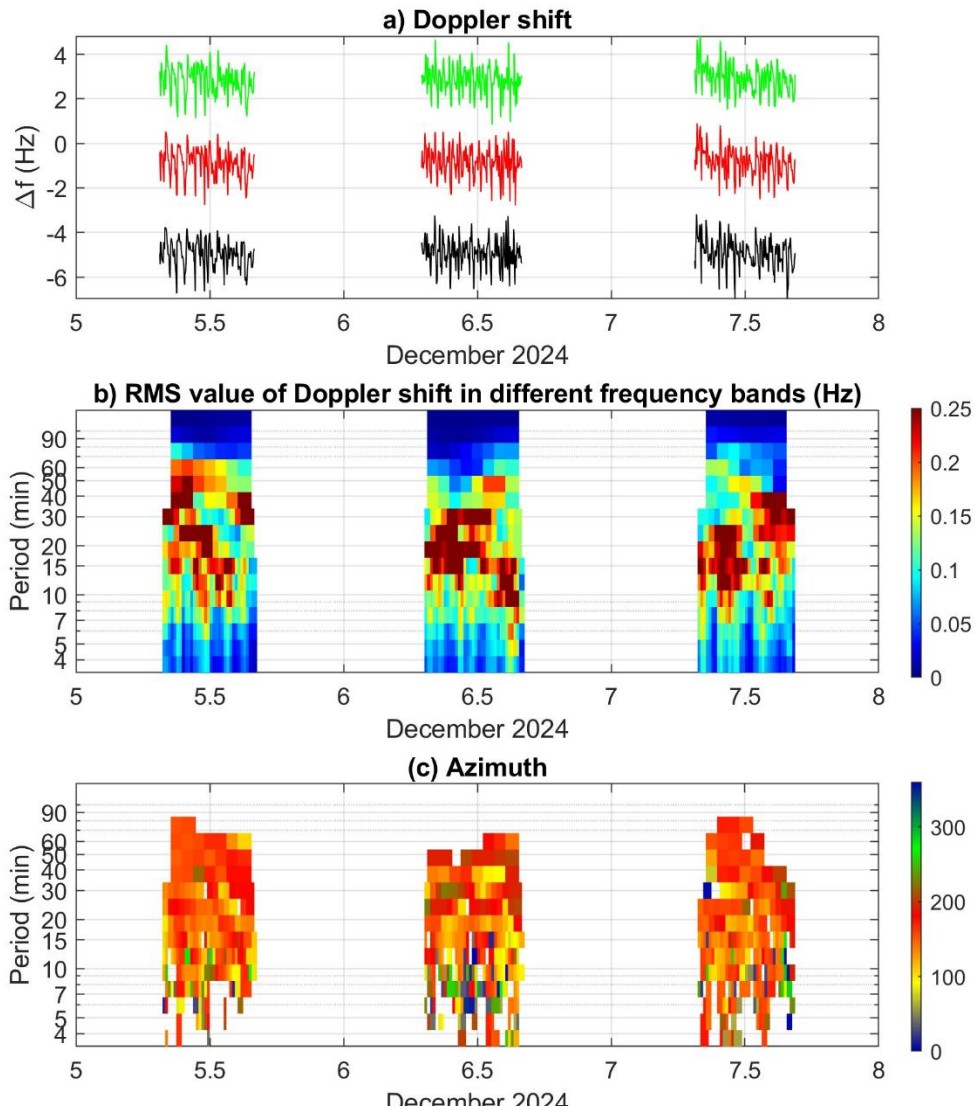

**Figure 15:** **(a)** Doppler shift frequencies of spectral density maxima for individual transmitter-receiver pairs (including artificial offsets) from X to Y. **(b)** Dynamic spectra (periodograms) of Doppler shift signals and **(c)** the propagation azimuth of waves, displayed as function of period and time for December 5-7, 2024.

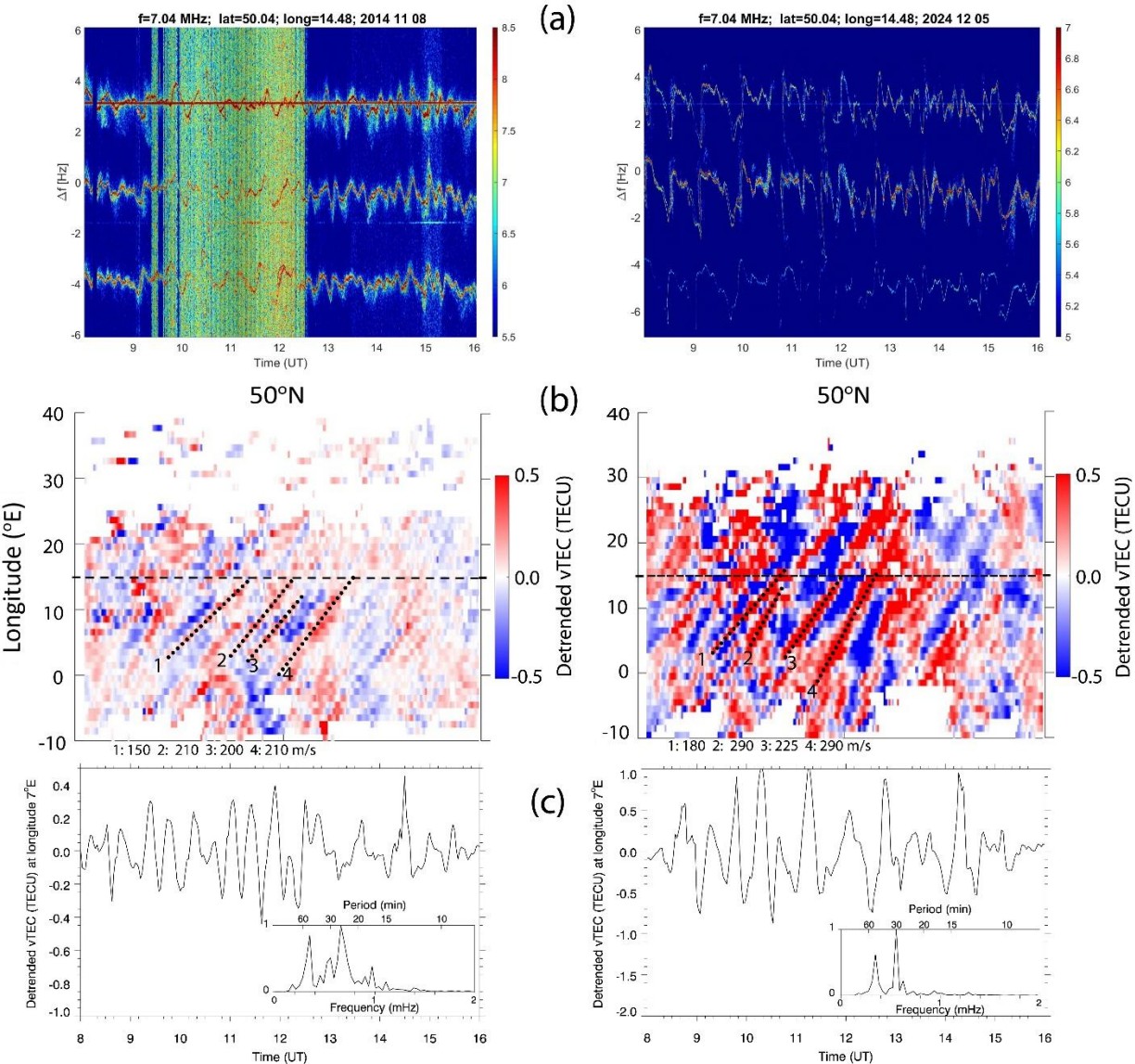

1187
**Figure 16: (a)** The Doppler shift spectrogram recorded at frequency 7.04 MHz on November 8,
2014, and December 5, 2024. **(b)** The detrended vTEC mapped along latitude of 50°N. The
dashed line shows the longitude of Prague. The approximate eastward velocities of TIDs shown
in dotted lines are estimated. **(c)** The detrended vTEC time series at longitude of 7°E and the
normalized FFT spectra.

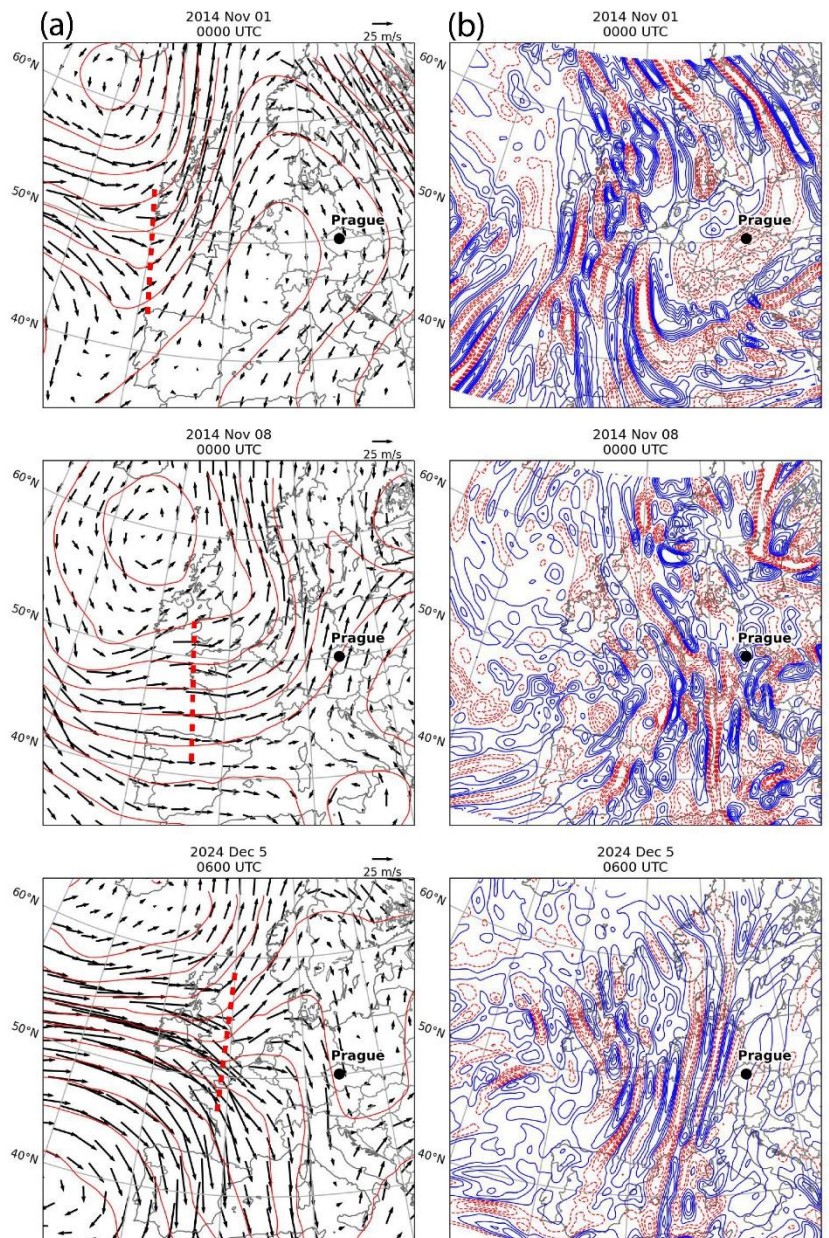

**Figure 17: (a)** The ERA5 geopotential height (red contours at intervals of 100 m), horizontal
winds (m/s) at 300-hPa level, with a probable source region of gravity waves indicated by red
dashed line. **(b)** The ERA5 divergence (positive in solid blue line) of the horizontal wind at 150-
hPa level, on November 1, November 8, 2014, and December 5, 2024..