# Peer review of "Observations of traveling ionospheric disturbances driven by gravity waves from sources in the upper and lower atmosphere"

_Annales Geophysicae, 2024_

## Author Response (AR1)

**Revisions in response to RC1**: , Anonymous Referee #1, 17 Jul 2024

The topic is important, and some interesting cases were chosen. However, there are several incorrect assumptions made. Most importantly, the assumption that equatorward propagating TIDs are generated by geomagnetic sources is not in line with peer reviewed literature and existing scientific knowledge regarding GWs/corresponding TIDs.

*Reply 1: We disagree. What does the reviewer imply by this statement? In Introduction we have sufficiently referenced papers back to 1960s and 1970s that demonstrated equatorward propagation of aurorally generated GWs/TIDs. For example, review by Hocke (1996) already discussed it in great length. This is also in line with our most recent paper (Prikryl et al., 2022) and many papers referenced there. Yes, recently, it was shown that polar vortex generates GWs that can also propagate equatorward, and we referenced paper by Frissell et al. (2016). However, this does not replace all the extensive previous work on GWs/TIDs from auroral sources, which in this paper we show can be further poleward than any ground magnetometers that could detect the geomagnetic activity associated with ionospheric currents. In the revised manuscript we included further references, including works on polar vortex generated GWs/TIDs.*

Furthermore, only a cursory analysis was provided for several cases. In some instances, arrows do not correspond to events that are appropriately matched in time, or arrows point to data that does not demonstrate anything.

*Reply 2: Of course, the arrows in Fig. 2 do not show exact correspondence between the IMF clock angle and TIDs. The IMF observed by ACE does not represent exactly the IMF impacting the dayside magnetopause. It would require at least a spacecraft in front of the bow shock to monitor the IMF (e.g., Prikryl et al. 2002). But the observations of pulsed ionospheric flows and corresponding TIDs provide sufficient evidence that points to sources of these TIDs in the high-latitude ionosphere. Such analysis that the reviewer chooses to call "cursory", provides clear evidence for sources of equatorward propagation TIDs (Prikryl et al., 2022). Furthermore, Figure 2d now includes FFT spectra of the IMF Bz, By, and the radar ground scatter power. The spectra of the IMF Bz and the Prince George radar ground scatter power (TIDs) are very similar providing further support for the coupling of solar wind to dayside magnetosphere as the source of the observed TIDs.*

Additionally, improper use of citations is present throughout the manuscript. Given these issues, I cannot recommend this manuscript for publication.

*Reply 3: Although citations of conference papers are not improper, and the AGU provides the citation form for the conference presentations, we removed all these references in the revised manuscript. The authors of these presentations have not published their results.*

In terms of lack of rigor in the analysis, the authors do not do a comprehensive lower atmosphere analysis for Jan 8 as was done for the Nov 2014 case (a brief mention of a figure for Jan 8 is included in the supplemental material). That said, the interpretation of lower atmospheric data is problematic for the Nov 2014 case. Dotted lines are drawn to denote regions of GW generation, yet ERA 5 shows widespread GWs in the troposphere. No analysis has been performed for GWs propagation in the stratosphere (this would be difficult to assess with ERA5). However, during times of the polar vortex, upper stratospheric conditions play a significant role in GW filtering, generation, and coupling into the thermosphere.

*Reply 4: There is no need for a lower atmosphere analysis for January 8, 2013, because the TID sources are shown to be in the upper atmosphere driven by solar wind coupling (see, Reply 2). For the November 2014 cases, the eastward propagating TIDs are associated with intensifying extratropical cyclones, and we pointed to likely process generating the GWs by referring to works by Uccellini and Koch (1987), Koch and Dorian (1988), and Plougonven and Zhang (2014). We used ERA5 reanalysis data for the 150-hPa level to support it, similarly to what the latter authors have done. While the whole atmosphere modeling could be used to further assess the circumstances at play, such modeling is beyond the scope of the present study, and we believe that the use of an existing peer-reviewed approach is sufficient at this time.*

In terms of citations, the manuscript cites several conference abstracts multiple times, in some cases appearing to disagree with the presenters. However, there is no publicly available link to the corresponding talks. Most concerning,

peer reviewed publications associated with/building off the cited conference talks were NOT cited. For example, this manuscript attempts to demonstrate that TIDs observed on Jan 8-9 were from geomagnetic sources, cites Bossert et al., 2021 multiple times (an AGU conference abstract, no publicly available talk) to state that Bossert et al said the TIDs are from the polar vortex, but does not give any greater context of the authors' findings. Yet, Bossert et al., 2022 has already demonstrated quiet-time/low Kp index TIDs that were likely generated by geomagnetic sources, and this work is not acknowledged. Similarly, a Becker et al EGU conference abstract is cited, but no Becker papers are referenced in the manuscript. This lack of proper citation of the relevant literature is unacceptable for a scientific paper.

Generally, the manuscript lacks adequate citations of current research regarding GW propagation and multi-step coupling from the lower to upper atmosphere, yet attempts to demonstrate one such observation of TIDs from the lower atmosphere on this topic (e.g. the Nov 2014 case).

*Reply 5: We removed all references to conference abstracts. References to more recent publication of GWs generated by polar vortex are added and discussed in the revised manuscript.*

Section 3.1: There is a somewhat cursory approach used here to justify that the TIDs originated from solar wind forcing. It is stated on lines 208-210 *"The clock angle controls the reconnection rate at the magnetopause (Milan et al., 2012). The TIDs can be approximately associated with the southward IMF turnings (positive deflections of the clock angle values towards 180 degrees marked by arrows in Fig. 2)."*

First, is it only the clock angle? Or is it also IMF? Is there a threshold at which $B_z$ or $B_y$ need to be perturbed in order for a TID to form? The perturbations are only a few nT at their strongest. Is that significant? Is there a modeling study that could be cited?

*Reply 6: The clock angle is the IMF. Yes, the IMF perturbations of a few nT are sufficient to modulate the magnetic reconnection, resulting in PIFs/TIDs (e.g., Prikryl et al., 2002, 2005). Further evidence for the January 2013 event is now provided (see, Reply 2).*

Will the TIDs only form at a specific location or are they expected to form all over the polar cap and auroral oval? The ACE data could be applied to quite a range of areas in the auroral region, but localized effects can vary drastically.

*Reply 7:  All these TIDs were generated by dayside sources of GWs in the cusp (ionospheric footprint of the magnetospheric cusp).*

It would appear that Figure 2 has hand drawn arrows pointing from the power fluctuations observed in PGR to the ACE clock angle plot. These arrows are not necessarily correlated. Some are drawn directly upwards, some are drawn at a slant in order to match some of the power fluctuations with perceived spikes in the ACE data. Some of the arrows (e.g. one drawn just after 22UT) doesn't even appear to correspond to any fluctuation in the IMF clock angle at all. Some of the larger power fluctuations have an arrow pointing to a very tiny change in the IMF clock angle. There appears to be no obvious correlation here, and no quantitative analysis was performed. This is not enough to demonstrate a link between TIDs present and geomagnetic activity.

Furthermore, there isn't a baseline for comparison of just how strong the ACE perturbations need to be, and what sort of TIDs and corresponding amplitudes would be generated. Prikyl 2022 is cited, but again, the data shown there are overplotting ACE data, and there is not a quantitative analysis performed. That case seemed to have a better match at least with the perturbations. So, it is difficult to compare the case presented in this paper with hand drawn arrows. Are there no other datasets that can be used to "provide evidence the observed TIDs could have originated from the magnetosphere/solar wind forcing" as is mentioned in line 212?

*See Reply 2. We are not demonstrating a link between the observed TIDs and geomagnetic activity (Kp index or ground magnetic field), rather, fluctuations in the IMF modulating ionospheric flows, which are observed by the King Salmon radar. Regarding the arrows, they are drawn to point to clock angle deflections towards 180°. Sure, some of them are subtle but as we pointed out above ACE does not necessarily observe exactly the IMF impacting the magnetopause. Unfortunately, there are few spacecraft in the upstream solar wind and none in the magnetosheath.*

An additional question would be whether the TIDs that are assumed to be associated with southward IMF turnings have a specific mechanism of generation? Are these related to gravity waves in any way? What is the expectation for TIDs being generated through "magnetosphere/solar wind forcing" (discussed on lines 2012-2013)? The author cites an AGU talk from 2021. While there does not appear to be a readily available copy online, was part of the Bossert et al 2021 argument that the observed TIDs/TADs on this date followed gravity wave polarization relations? Would this also be expected for cases of TIDs originating from magnetosphere/solar wind forcing? The authors will need to provide adequate references on this and more detail describing the mechanism of TID generation as well as the specific location with regards to ionospheric convection/currents. How close does this need to be to the region of observed TIDs?

*Reply 8: The mechanisms for energy transfer to the thermosphere can be Joule heating, precipitation, or simply ion drag (Lorentz force) by swings in convection (PIFs) generating GWs. Solar wind modulation of cusp particle signatures was associated with ionospheric flows (Rae et al., 2004). This reference is now included in the revised manuscript. These authors also concluded that solar wind Alfvén waves modulate magnetopause reconnection and referenced some of the works by Prikryl et al. that supported it. Subsequent works (e.g. Prikryl et al. 2005) showed that such events are also sources of GWs that modulate ionospheric densities and drive equatorward propagating TIDs.*

Regarding section 3.2:

Figure 5: Again, magnetometer perturbations are shown plotted above radar power fluctuations and LOS velocities. There is no context for geographic location. One single magnetometer is used. It is also not clear from the plots that the magnetometer perturbations correspond to fluctuations observed in the radar power or LOS velocities.

The text bounces back to figure 4: *"For the first event, Figs/ 4b and 4c show the FFT spectra of detrended time series of IMF Bz, solar velocity Vz, and NAL X-component, and the Hankasalmi radar ground scatter power displaying peaks at similar frequencies/periods."*

It is not surprising that the ACE data (Bz and Vz) are showing similar periods. The NAL X and radar are an interesting comparison, but this needs to be discussed more. The only information given is "beam 11, gate 25, slant range 1305km)" is being used. Why just one altitude? Why just one beam? Looking at Figure 5b, the previous statement seems a bit of an incomplete story as to what is going on. There are clear perturbations in the power near 1305km from 8-11UT, but not necessarily perturbations in the NAL X magnetometer data. Then, from 12-16UT, there are shorter period perturbations, and these do not appear to correlate with NAL X perturbations during 12-16UT. So, the analysis used does not seem to be appropriately describing the data.

*Reply 9: ACE data (Bz and Vz) are showing similar periods because the fluctuations are Alfvén waves. This is not always the case. Regarding magnetometer data, all IMAGE magnetometers are used to estimate 1D equivalent ionospheric currents that are now included in Figures 5 and 6. Similar periods are evidence that the solar wind Alfven waves modulated the PIFs, associated currents and TIDs. The choice of the range midway the ground scatter bands (TIDs) sufficiently represent the TIDs. Other choices would yield similar results.*

Line 247-248: "The equatorward TIDs were observed at least down to latitude of 50N."

How was the propagation direction determined in this case?

*Reply 10: Animations of TEC maps showing equatorward propagation of TIDs are provided in the Supplement material.*

-The data presented in section 3.2 (e.g. Fig 9 b) may be more in line with what has been observed with LSTIDs and aurora (e.g. Zhang et al., 2019 **https://doi.org/10.1029/2019JA026585**). That said, further discussion regarding how these are classified as LSTIDs (what is the horizontal wavelength and the period) should be included. Also, a discussion of how the propagation direction was determined needs to be included.

*"In summary, the cases discussed in sections 3.1 and 3.2 highlight the importance of solar wind coupling to the MIT system, particularly on the dayside, in the generation of AGWs/TIDs."* Lines 282-283.

Section 3.1 has not provided adequate analysis to justify this statement, though the notion is important for further discussion. Section 3.2 presents a case similar to those which have previously been presented for LSTIDs generated from aurora.

**Reply 11:** *PIFs and ground magnetic field fluctuations are all observed in the ionospheric cusp in the noon sector and are followed by TIDs. This is sufficient evidence they are the result of solar wind-MIT coupling and further evidence is now provided (see, Reply 2). TIDs with wavelength greater than 1000 km are classified as LSTIDs (Hunsucker (1982).*

Section 4

*Line 292: "At high latitudes, they propagate predominately equatorward suggesting likely auroral sources."* This is not scientifically accurate or in line with gravity wave theory. Studies have demonstrated GWs generated from lower altitudes can propagate equatorward (for example Becker et al., 2022 https://doi.org/10.1029/2022JA030866).

**Reply 12:** *Well, this study shows that the observed GW/TIDs not only can be generated by auroral sources, but that they did originate in the cusp (see, Reply 8-9) through direct coupling of solar wind to dayside magnetosphere. This is not the same as nightside auroral sources, which are due to impulsive electrojet intensifications, for example during auroral substorms. This sentence is amended.*

*Line 293 "At mid to low latitudes, the azimuth of MSTIDs varies suggesting sources in the troposphere."* Again, there is no scientific basis for this statement. It also doesn't suggest tropospheric sources over other atmospheric sources.

**Reply 13:** *The presence of an intensifying extratropical cyclone just west of the observed south-eastward propagating MSTIDs suggests possible convective sources in the troposphere (e.g., Azeem et al., 2018).*

Lines 296-300: This section needs clarification, specifically the statement "with southeastward propagation more likely in winter months, suggesting that cold season low pressure systems in the northeast Atlantic are sources of the GWs." Is this inferred for the current research or cited research from Chum et la., 2021. Chum et al., 2021 states the higher likelihood of southeast propagation in the winter months, but does not make the inferred connection to low pressure systems in the Atlantic. In that case, GWs in the ionosphere were found to propagate against background winds.

**Reply 14:** *This can be inferred from the cited research by Chum et la., 2021. But the eastward propagation is a clear indication the sources are not auroral. Presence of intensifying extratropical cyclones suggests a tropospheric source.*

*Lines 309-320 (and Figs 11 and 12):* Please provide more information about the analysis performed here. What are the errors associated with the retrieved periods and propagation azimuths. Is a particular analysis method being followed? Please provide a reference here.

**Reply 15:** *Chum et la., 2021*.

*Line 325-327: "During the period from November 1 to 8, 2014, we distinguish between aurorally-generated TIDs propagating equatorward from high latitudes (Section 3.2) and south-eastward propagating MSTIDs at mid latitudes by observed origin location."*

Again, this method cannot actually determine whether a wave is aurorally generated. Any wave generated in the lower atmosphere can propagate equatorward. This is especially true for gravity waves in the presence of the diurnal tide on the dayside.

**Reply 16:** *As we already stated in the above replies, we show evidence of TIDs propagating equatorward from sources in the cusp.*

*Line 328-330: "Low-pressure systems deepening over the Northeast Atlantic shown in the surface pressure analysis charts, were likely sources of MSTIDs propagating eastward to southeastward."*

It is certainly possible that this is a source of GWs that relate to MSTIDs. However, there is no other analysis provided here that would further justify this assertion.

*Reply 17:* *Provided in Section 4.2*

*Line 332-333: "At the same time, the vTEC maps on both days also reveal equatorward propagating TIDs at latitudes down to 50N that originated in the cusp ionospheric footprint"*

Again, "equatorward propagating" does not definitively prove TIDs originating from auroral regions.

*Reply 18:* *see, Replies 8, 9 and 12*

Regarding Fig 13 a,b and Lines 332-334: *"At the same time, the vTEC maps on both days also reveal equatorward propagating TIDs at latitudes down to ~50N that originated in the cusp ionospheric footprint over Svalbard, as already discussed in Section 3.2."*

While section 3.2 provided data from Nov 1 and Nov 5, no analysis was performed for Nov 5. For Nov 1, the NAL X perturbations do not necessarily correspond to the perturbations in the radar power, and no adequate description of the formation of GW-like TID features has been provided. While it is possible the TIDs are linked to geomagnetic activity on this day (Nov 5), this is a hypothesis at this point. The datasets shown in 3.2 certainly don't prove that TIDs on different days, such as Nov 8, originated near the cusp. What is concerning about this instance on Nov 8, is that there is a notable polar vortex globally, which can be a source of GWs in the stratosphere (see publicly available MERRA2 data, demonstrates the polar vortex and strong wind shears in the stratosphere at the beginning of November). Additionally, there are now studies demonstrating GW generation from the northern side of the polar vortex and propagation across the polar region (e.g. see Becker et al, 2022 doi:10.1029/2022JA030866 for example Fig 6e,f), which results in TIDs/TADs from GWs in the thermosphere.

*Reply 19:* *For Nov 5, Figure 6b now includes equivalent ionospheric currents. The FFT analysis for Nov 5 can be now viewed in the Supplement Figure S5. The onset of intensifications of eastward electrojet clearly coincided with the onset of PIFs followed by TIDs. While the correspondence between the current intensifications and PIFs/TIDs is only approximate because it is subject to limitations/uncertainty of HF propagation, there is no doubt about the source of GWs in the cusp and thus the coupling of solar wind to dayside magnetosphere, as discussed in the above replies and in the manuscript.*

Section 4.2:

This is interesting discussion. However, a quantitative analysis is again lacking. In Fig 15, was the axis of inflection drawn on, or was this calculated? How was this determined? More references are needed here to back this up. Additionally, one can look at satellite data for this time period and see that GWs are all over the stratosphere around the region of the polar vortex and not just at the dotted line region drawn in Fig 15a. The divergence of the horizontal wind at 150-hPa can certainly give an idea of where GWs are present in ERA5 output, and in the cases presented, they are quite widespread. It is also odd that Nov 8 ERA data are used to try and describe TIDs observed on Nov 8. It would take GWs some period of time to propagate from the troposphere to the thermosphere. 11 hours is on the shorter side of assumptions for slow moving GWs generated from jets. It would also be appropriate to use data from Nov 7 as well.

*Reply 20:* *Yes, the axis of inflection was drawn on, and as stated, it is approximate. We are simply referring to, and using, the conceptual model pointing to a possible source of GWs. Yes, these GWs need to propagate to thermosphere. The ERA5 reanalysis showing divergence at 150 hPa indicating GWs in the lower stratosphere is for 0 UT. There was enough time for these GWs to propagate to the ionosphere, where TIDs started to be observe at least 10 hours later. Data for Nov 7 are shown in Supplementary material.*

*Lines 390-397: "Bossert et al. (2021) observed GWs generated by stratospheric vortex. There was an extratropical cyclone intensifying just south-west of Alaska. Using the ERA5 reanalysis, similar to Figs. 15e,f, north-eastward propagating GWs in the stratosphere are found (Fig. S7 in the Supplement) but no corresponding TIDs can be resolved in the detrended TEC maps, possibly because of sparce coverage by GNSS receivers. However, mesoscale GWs propagating eastward and upward into the stratosphere generated by geostrophic adjustment processes and shear instability may be common and could be driving MSTIDs."*

Unfortunately, this is not a thorough investigation. Further instrumentation would need to be included to assess GWs in the stratosphere, as ERA5 does not necessarily capture dynamics on the top side of the polar vortex in the upper stratosphere/mesosphere. A quick check online from AIRS Jan 8 and 9 2013 (https://datapub.fz-juelich.de/slcs/airs/gravity_waves/html/view_2013_009.html) also shows significant poleward GW activity in the stratosphere north of AK, and these breaking GWs could produce secondary GWs that propagate equatorward.

*Reply 21:  Thank you. That's great that the GWs in the lower stratosphere that can be seen in the ERA5 reanalysis were also seen by AIRS (https://datapub.fz-juelich.de/slcs/airs/gravity_waves/html/view_2013_008.html).*

[Figure]

AIRS | 2013-01-08, 13:30 LT

*But even if they produced secondary GWs propagating equatorward, which is just a hypothesis, they could not be related to TIDs observed by PFISR and SuperDARN that had a source in the cusp, as shown in the manuscript. Furthermore, the next and the subsequent day, when equatorward propagating TIDs were again observed by PFISR and SuperDARN radars, such poleward GW activity in the stratosphere over Alaska did not seem to be present.*

Again, a conference talk is being cited, so it is hard to tell what is being referenced here. That said, the plots provided in the manuscript have not provided definitive evidence that the TIDs are indeed geomagnetically induced as opposed to driven from the lower atmosphere.

*Reply 22:  As already stated in Reply 3, we deleted the references to conference papers. We disagree with the reviewer that plots in Figures 4-10 do not provide definite evidence. They do, and some of the Figures have been updated to include EICs to support it. We do not have evidence that these TIDs were driven from the lower atmosphere, for example, from browsing AIRS online (https://datapub.fz-juelich.de/slcs/airs/gravity_waves/html/view_2014_305.html).*

Finally, I will note here that the citations of conference talks are not appropriate. Especially considering the talks themselves are not publicly available. The authors and coauthors cited, Becker et al, and Bossert et al, have published work in recent years that likely builds on previous conference presentations. These citations should be included in this paper. Interestingly, the work here builds off what has been written in Becker et al., 2022, Vadas et al., 2022, and Bossert et al., 2022. Becker et al., 2022 (https://doi.org/10.1029/2021JD035018 ) discusses methods for assessing vortex generated GWs from model output. It is the combination of MKS and MPS as calculated by a model that determine the regions of jet generated GWs. Additionally, this paper shows examples of over pole GW propagation (e.g. see Figure 3a). Vadas et al., 2022 discussed observations of polar vortex generated GWs and subsequent secondary GW generation in the polar region (https://doi.org/10.1029/2022JD036985). Bossert et al., 2022 (https://doi.org/10.1029/2022GL099901) discussed a strong TID/TAD event observed during an SSW (a few days after the Jan 2013 date discussed here) and suggested that the large-scale TID/TAD was related to geomagnetic activity despite a low Kp. This is not a new concept, though it is important to furthering the discussion of waves and variability in the thermosphere.

So, a more thorough discussion and consideration of current work should have been addressed in this manuscript.

***Reply 23:*** *Again, we removed the conference references. We have now acknowledged and referenced all the above works but to undertake such approach is beyond the scope of this study.*

---

## Referee Report (RR1)

**Title:** Observations of traveling ionospheric disturbances driven by gravity waves from sources in the upper and lower atmosphere.

**Overview of Manuscript**

This work investigates the sources of ionospheric disturbances due to atmospheric gravity waves due to high latitude space weather and low latitude tropospheric dynamics. They used multiple observational techniques to detect traveling ionospheric disturbances (TIDs) with large and medium scale characteristics, and atmospheric gravity waves (AGWs). Using observations and reanalysis data, they investigated the upper atmospheric dynamics that possibly excited the large and medium scale TIDs originating from the troposphere. The manuscript has the potential to contribute significantly to literature; however, there are some issues that need to be addressed.

It is concerning that no relationship between the observed wave and the source has been made in the current state of the manuscript. Instead, only what appeared to be a literature review of the previous works was done in conjunction with the results presented. Additionally, the presentation of the work from abstract to the conclusion is not in any chronological order, thus making it very difficult to comprehend the work and make the necessary relationship between any observed phenomenon and the other. For instance, the author mentioned the observed AGWs that induced MSTIDs. No comparison was made in order to prove that the AGWs indeed induced the observed MSTIDs. Also, they mentioned that the observed AGWs were excited though space weather event or tropospheric activity without arguing with evidence whether the said source is really the source or not. They provided some evidence and in some cases links to the evidence. Most of the links are not working.

Considering this lack of details, and the other comments listed below, I suggest the manuscript be subjected to a major revision. If these issues can be addressed, and the techniques more properly explained, the manuscript will contribute significantly to literature.

**Major Comments**

- **Abstract:**
  1. The abstract did not capture the necessary aspects of the work to give the reader a complete idea about the work. The work was not introduced, and the order of presentation makes it difficult follow. Kindly revise the abstract.

- **Introduction:**
  1. The introduction of the work is too precise and lacks a chronological presentation. There is no detailed presentation in this section. I encourage the authors to revise the introduction.

2. No in-depth and related literature review has been made.

- **Data sources and methods**

1. The data sources and methods are not detailed enough to understand the step by step process to analyze the data. They are possibly assuming the readers are familiar with the subject. They need to expand and give more information on the methodology.

2. Kindly provide a map showing the location where each data was collected. The comprehension as to the location where the authors are referring to in the manuscript is confusing, it would be better for the reader to know the locations of data collection.

3. The authors gave shallow descriptions of the methods used in retrieving the parameters. I suggest they restructure this section into subsections for each instruments and give a detail description of each instrument as well as the methodology employed in retrieving the parameters. The section can be structured as follows:

*2. Data sources and methods*

*2.1. Advanced Modular Incoherent Scatter Radar (AMISR)*

Description of the instrument and the methodology/ data analysis

*2.2. Multi-point and multi-frequency continuous HF Doppler sounding system*

Description of the instrument and the methodology/ data analysis

*2.3. SuperDARN*

Description of the instrument and the methodology/ data analysis

*2.4. Global Navigation Satellite System (GNSS)*

1. Description of the instrument and the methodology/ data analysis

- **Result**

The presentation of the result is quite confusing. This section also needs further revision.

- **AGWs/TIDs originating from sources in the troposphere**

This section needs total revision with more graphical evidence.

- **Discussion**

1. The discussion and conclusion are too shallow and fail to mention the main scientific contribution of the work to literature.

- **Summary and conclusions**

    1. The conclusion does not reflect the results presented and discussed. The authors intend to investigate the source of the AGWs induced MSTIDs. However, this has not been demonstrated in the current state of the manuscript.

**Minor Comments**

- **Abstract:**
    1. For instance, Between **Lines 21-24**: The work was not introduced properly, rather information about the instruments were given.

    3. **Line 27-29:** they mention the aim of the work, however, the preceding and succeeding sentences are not compatible, making the flow in the write-up interruptive. Kindly revise.

- **Introduction:**

    1. **Line 47-48:** *TIDs generated by AGWs originating in the lower atmosphere come from a variety of sources… .*
        a. The authors need to be mindful of the choice of words. AGWs capable of propagating to the ionosphere and modulating TEC are considered TIDs or perturbation generated in the ionosphere. It will be better to say TIDs are GWs modulated TEC or better still to say TIDs are driven by GWs. This is the case when considering TIDs mostly originating from the lower atmosphere.
        b. This sentence is too long. I suggest you break it into two parts:
            i. sources at the lower latitude and equatorial regions.
            ii. sources at high latitude.

2. **Line 59-62:** *The Joule heating due to the ionospheric currents of in the lower thermosphere is a source of equatorward propagating AGWs … .*
    a. It should rather be "*The Joule heating due to the ionospheric currents in the lower thermosphere is a source of equatorward propagating AGWs …. .* [remove the "**of**"]

- **Data sources and methods**

    1. **Line 83-84:** *To retrieve TIDs, background densities are removed by applying Savitzky-Golay filter (Press and Teukolsky, 1990).*

        a. How was the data preprocessed before the application of the Savitzky-Golay filter? Why is this filtering method chosen over the other methods?

    2. **Line 86-98:** The authors just cited the works done by others and possibly assumed the readers are familiar with the method. It is important to state and discuss the specific methods used, even if the readers are familiar.

    3. **Lines 100-114:** a brief but yet no detailed description of the SuperDARN is given. However, similar to the other instruments, basically no information on the methodology used to first preprocess the data, followed by the data analyzing to retrieve the necessary background information or wave parameters. Kindly provide this information.

    4. **Line 116:** What is SECS inversion technique? Why have you chosen this approach over the others? Details are needed to enhance the understanding of the reader.

    5. **Line 123-136:** Similar to other comments on the previous instruments.

    6. **Line 134-135:** The authors need to give enough details on this procedure instead of citing reference.

    7. **Line 138-142:** This section is not supposed to be here. It should be in the acknowledgement.

- **Result**

    o **AGWs/TIDs originating from lower thermosphere at high latitudes**

1. **Line 146-167:** This section needs to be in the introduction.

- **Event of January 8/9, 2013**

The presentation of this section:

2. the presentation is not arrange such that the reader can easily understand.

3. the presentation of the corresponding Figures is not in sequential order. For instance, in some part of the text, results presented in Figure 9 are presented before Figure 8.

4. some undefined abbreviations are found within this section.

- **Events of November 1 and 4-5, 2014**

5. What are the parameters of the observed waves? How do you know they are large-scale characteristics? This section needs to be revised. Either a table or a plot needs to be provided with the wave parameters. Putting some of them in the texts is not enough.

1. **Line 301-302:** *Atlantic are sources of the GWs, which supports previously published results referenced above and points to winter jet stream as a likely source of GWs*.

    a. Please cite some references.

- **Events of November 1-8, 2014**

2. **Line 376-378: …** *were likely sources of MSTIDs propagating eastward to southeastward, as observed in the detrended vTEC maps (indicated by arrows in Figs. 13a,b) on November 1 and 8, 2014.*

    a. It has been mentioned that the propagation of the MSTIDs are indicated by arrows in Figure 13a.b. However, no arrows have been plotted. Only ">>" were used. Kindly use real arrows.

3. **Line 388-390:** *As described in more detail by Chum and Podolská (2018) and Chum et al. (2021), the use of well correlated signals at two or three different frequencies makes it possible to determine a 3-D phase velocity vector.*

a. How will the reader know the details in Chum and Podolská (2018) and Chum et al. (2021) with respect to the obtained result? Kindly mention here the exact point of these references.

- **Physical mechanism of GW generation in the troposphere**

4. **Line 439:** *Using the ERA5 reanalysis, similar to Figs. 15e,f, north-eastward propagating GWs in the ….*

a. This is a bit confusing. There is no Figs. 15e,f. Kindly check and correct.

- **Discussion**

1. **Line 454-455:** "*In Section 3.1, we have shown evidence that even during a geomagnetically very quiet period the TIDs that were observed by PFISR in Alaska can be attributed to sources at high latitudes*".

a. This aspect of the manuscript in Section 3.1, appear more of literature review and presentation of result. I would like to encourage the authors to really show (with diagrams) other evidence that the high latitude sources were really the possible sources of the detected TIDs.

---

## Referee Report (RR2)

**Title:** Observations of traveling ionospheric disturbances driven by gravity waves from sources in the upper and lower atmosphere.

**Overview of Manuscript**

This work presents Observations of traveling ionospheric disturbances driven by gravity waves from sources in the upper and lower atmosphere. They use a multi-instrument approach with the aim of attributing observed TIDs to atmospheric gravity waves generated in the lower thermosphere at midlatitudes. The work has the potential to contribute to existing literature if revised and some issues fixed.

I therefore recommend the manuscript be accepted after revision.

**Comments**

1. The authors did not capture the important aspects of the work done. Please rewrite the abstract so it captures the attention of the reader and gives a clear overview of the work.

2. The introduction is not very clear. It is very confusing to read and is more of a literature review than an introduction of the subject matter. Please rewrite it.

3. The authors keep citing and referring the reader to works done without stating exactly in the text what they want the reader to know. Please cite and state the point you want to let the readers know in the text.

4. Please elaborate on the SECS inversion technique and it´s advantage over other techniques.

5. Please can the authors represent the instruments used and also the location of the studies in a table or map? This will help the reader.

6. The methods used to arrive at the results are very shallow and would be difficult for replication by the reader. The authors should please give detailed and step-by-step write-up of the methodology employed.

7. Please the authors should state clearly in the text the contribution (the new findings) of this work to already existing literature.

---

## Referee Report (RR3)

**Title:** Observations of traveling ionospheric disturbances driven by gravity waves from sources in the upper and lower atmosphere.

**Overview of Manuscript**

This work investigates the sources of ionospheric disturbances due to atmospheric gravity waves due to high latitude space weather and low latitude tropospheric dynamics. They used multiple observational techniques to detect traveling ionospheric disturbances (TIDs) with large and medium scale characteristics, and atmospheric gravity waves (AGWs). Using observations and reanalysis data, they investigated the upper atmospheric dynamics that possibly excited the large and medium scale TIDs originating from the troposphere. The manuscript has the potential to contribute significantly to literature; however, these minor issues need to be addressed.

The authors have responded well to comments and implemented satisfactorily my comments and suggestions. However, there are some minor issues that needs to be solved, especially in the abstract and data sources and methods sections. In the case of the abstract, I will recommend that the authors consider rewritting it entirely with concise but direct and needed details.

The manuscript in the current needs minor revision after which it can be accepted for publication.

**Abstract:**

**General Comment.**

The abstract is quite confusing. This, I consider to come from the presentation of instruments used and their respective methodologies. Please, carefully but in simple terms consider writting the instrumentation and methodology aspects. Also, capture in clear terms the major contribution of the work.

**Minor Comment(s):**

1. **Lines 22 and 24:** Kindly consider rephrasing the sentence: " .... generate gravity waves driving equatorward propagating medium- to large-scale traveling ionospheric disturbances (TIDs) ... and ..... extratropical cyclones are sources of gravity waves driving medium-scale TIDs. The phrase "generate gravity waves driving" .... is a bit confusing.

**Data sources and methods**

**General Comment(s):**

The authors have implemented satisfactorily in relation to this section. However, I would like to suggest to them to iclude the map of the GNSS reciever stations in the supplementary materials to the Fig.1 here (in the main manuscript). Thus, making Fig.1 to comprise of panels (a) – for the current fig. 1, (b) – for the first local domain, and (c) – for the second local domain.

The legends are not explained in the text. Infact no legend were defined in the figures for the maps (both the one in the main manuscipt and in the supplementary material). Kindly update them and explain them in the text according to the paragraph in which each instrument was described.

**Minor Comment:**

1. **Line 24:** The GNSS data for this ???? . …. The sentence seems incomplete, kindly revise.

---

## Referee Report (RR4)

**Title:** Observations of traveling ionospheric disturbances driven by gravity waves from sources in the upper and lower atmosphere.

**Overview of Manuscript**

This work presents Observations of traveling ionospheric disturbances driven by gravity waves from sources in the upper and lower atmosphere. They use a multi-instrument approach with the aim of attributing observed TIDs to atmospheric gravity waves generated in the lower thermosphere at midlatitudes. The work has the potential to contribute to existing literature if revised and some issues fixed.

I therefore recommend the manuscript be accepted after the implementation of the comment.

**Comments**

1. The authors should please further simplify the abstract capturing only the important aspects. Although they have worked on it considereing the previous version, they should be encouraged to go straight to the point.

---

## Author Response (AR2)

**Reply to Referee Report 1**

**Title:** Observations of traveling ionospheric disturbances driven by gravity waves from sources in the upper and lower atmosphere.

**Overview of Manuscript**

This work investigates the sources of ionospheric disturbances due to atmospheric gravity waves due to high latitude space weather and low latitude tropospheric dynamics. They used multiple observational techniques to detect traveling ionospheric disturbances (TIDs) with large and medium scale characteristics, and atmospheric gravity waves (AGWs). Using observations and reanalysis data, they investigated the upper atmospheric dynamics that possibly excited the large and medium scale TIDs originating from the troposphere. The manuscript has the potential to contribute significantly to literature; however, there are some issues that need to be addressed.

It is concerning that no relationship between the observed wave and the source has been made in the current state of the manuscript. Instead, only what appeared to be a literature review of the previous works was done in conjunction with the results presented. Additionally, the presentation of the work from abstract to the conclusion is not in any chronological order, thus making it very difficult to comprehend the work and make the necessary relationship between any observed phenomenon and the other. For instance, the author mentioned the observed AGWs that induced MSTIDs. No comparison was made in order to prove that the AGWs indeed induced the observed MSTIDs. Also, they mentioned that the observed AGWs were excited though space weather event or tropospheric activity without arguing with evidence whether the said source is really the source or not. They provided some evidence and in some cases links to the evidence. Most of the links are not working.

Considering this lack of details, and the other comments listed below, I suggest the manuscript be subjected to a major revision. If these issues can be addressed, and the techniques more properly explained, the manuscript will contribute significantly to literature.

**Major Comments**

- **Abstract:**
  1. The abstract did not capture the necessary aspects of the work to give the reader a complete idea about the work. The work was not introduced, and the order of presentation makes it difficult follow. Kindly revise the abstract.

*Reply. We have rewritten the abstract to better capture the aspects of this work, highlighting the multi-instrument techniques of TIDs observations and physical mechanisms that are sources of atmospheric gravity waves driving the TIDs.*

- **Introduction:**
  1. The introduction of the work is too precise and lacks a chronological presentation. There is no detailed presentation in this section. I encourage the authors to revise the introduction.

*Reply. The introduction has now been rewritten and reorganized. In addition to a literature review that is impossible to be made chronological, considering the multifaceted aspects of TID research, the content of the manuscript is introduced. The purpose of this work to show that gravity waves driving equatorward propagating TIDs at high latitudes are generated by solar wind – MIT coupling on the dayside, even during low geomagnetic activity. The eastward propagating TIDs waves are attributed to sources of gravity waves generated in the troposphere by geostrophic adjustment processes and shear instability. This physical mechanism has not been previously considered as a source of TIDs.*

  2. No in-depth and related literature review has been made.

*Reply. In the introduction, the referencing of the published papers is now organised in paragraphs, each dedicated to specific aspects of the previous work (i.e., theory of GWs and their relation to aurorally generated TIDs, lower atmospheric sources of GWs/TIDs (including polar vortex that has been recently, in our opinion, overemphasized), and TIDs detection techniques.*

- **Data sources and methods**

  1. The data sources and methods are not detailed enough to understand the step by step process to analyze the data. They are possibly assuming the readers are familiar with the subject. They need to expand and give more information on the methodology.

*Reply. Descriptions of the techniques and methodologies are now expanded and reorganized into subsections of Section 2.*

  2. Kindly provide a map showing the location where each data was collected. The comprehension as to the location where the authors are referring to in the manuscript is confusing, it would be better for the reader to know the locations of data collection.

*Reply. A map of instruments used in this study is now shown in the new Figure 1.*

  3. The authors gave shallow descriptions of the methods used in retrieving the parameters. I suggest they restructure this section into subsections for each instruments and give a detail description of each instrument as well as the methodology employed in retrieving the parameters. The section can be structured as follows:

  *2. Data sources and methods*

  *2.1. Advanced Modular Incoherent Scatter Radar (AMISR)*

*2.2. Multi-point and multi-frequency continuous HF Doppler sounding system*

Description of the instrument and the methodology/ data analysis

*2.3. SuperDARN*

Description of the instrument and the methodology/ data analysis

*2.4. Global Navigation Satellite System (GNSS)*

1. Description of the instrument and the methodology/ data analysis

***Reply.*** *Thank you for the suggestion to structure this section. Descriptions of the specific techniques and methodologies are now discussed in more detail, and the section is reorganized.*

- **Result**

The presentation of the result is quite confusing. This section also needs further revision.

***Reply.*** *We have now revised/reorganized and expanded Sections 3 and 4 discussing the specific results in more detail.*

o **AGWs/TIDs originating from sources in the troposphere**

This section needs total revision with more graphical evidence.

***Reply.*** *Section 4 has now been rewritten/restructured and additional data are presented to show more graphical evidence that is now discussed in more detail.*

- **Discussion**

1. The discussion and conclusion are too shallow and fail to mention the main scientific contribution of the work to literature.

***Reply.*** *We have now more clearly stated the main contribution of this work (i.e., solar wind – MIT coupling on the dayside generating TIDs, even during low geomagnetic activity; sources of gravity waves generated in the troposphere by geostrophic adjustment processes driving TIDs). The results are now presented in more detail in Sections 3 and 4.*

- **Summary and conclusions**

  1. The conclusion does not reflect the results presented and discussed. The authors intend to investigate the source of the AGWs induced MSTIDs. However, this has not been demonstrated in the current state of the manuscript.

*Reply. We do not have any means to observe AGWs in the upper neutral atmosphere, but Nykiel et al. (2024) (now referenced in Introduction) showed that Joule heating is a primary energy source for the night-time TIDs triggered in the auroral region, while the daytime TIDs can be also driven by precipitating particles in the polar cusp. PIFs are known to be associated with poleward moving auroral forms (precipitation), a source of AGWs/TIDs. All TID cases presented in Section 3.1 and 3.2 are shown to have sources at high latitudes. Fig. 3 discussed in Section 3.1 shows PIFs poleward of Alaska, which are the sources of AGWs/TIDs. Figs. 6 and 7 (in the European sector), and Fig. 8 (in the North American sector) show EICs, indicating the sources of TIDs. We believe this is now adequately summarized in Section 6.*

**Minor Comments**

- **Abstract:**
  1. For instance, Between **Lines 21-24**: The work was not introduced properly, rather information about the instruments were given.

*Reply. Abstract has been revised to introduce this work (see, our reply above).*

  3. **Line 27-29:** they mention the aim of the work, however, the preceding and succeeding sentences are not compatible, making the flow in the write-up interruptive. Kindly revise.

*Reply. The abstract has been revised and the sentences rewritten.*

- **Introduction:**

  1. **Line 47-48:** *TIDs generated by AGWs originating in the lower atmosphere come from a variety of sources… .*
     a. The authors need to be mindful of the choice of words. AGWs capable of propagating to the ionosphere and modulating TEC are considered TIDs or perturbation generated in the ionosphere. It will be better to say TIDs are GWs modulated TEC or better still to say TIDs are driven by GWs. This is the case when considering TIDs mostly originating from the lower atmosphere.
     b. This sentence is too long. I suggest you break it into two parts:
        i. sources at the lower latitude and equatorial regions.
        ii. sources at high latitude.

*Reply. The introduction has now been rewritten considering the referee's suggestions. Thank you.*

2. **Line 59-62:** *The Joule heating due to the ionospheric currents of in the lower thermosphere is a source of equatorward propagating AGWs … .*
   a. It should rather be "*The Joule heating due to the ionospheric currents in the lower thermosphere is a source of equatorward propagating AGWs …. .* [remove the "**of**"]

***Reply.*** *Corrected.*

- **Data sources and methods**

  1. **Line 83-84:** *To retrieve TIDs, background densities are removed by applying Savitzky-Golay filter (Press and Teukolsky, 1990).*

     a. How was the data preprocessed before the application of the Savitzky-Golay filter? Why is this filtering method chosen over the other methods?

***Reply.*** *This is now discussed in more detail in the expanded Section 2.1.*

  2. **Line 86-98:** The authors just cited the works done by others and possibly assumed the readers are familiar with the method. It is important to state and discuss the specific methods used, even if the readers are familiar.

***Reply.*** *This is now discussed in more detail in the revised Section 2.*

  3. **Lines 100-114:** a brief but yet no detailed description of the SuperDARN is given. However, similar to the other instruments, basically no information on the methodology used to first preprocess the data, followed by the data analyzing to retrieve the necessary background information or wave parameters. Kindly provide this information.

***Reply.*** *To address this, we have modified the text in lines 100-114 to provide a more thorough description of the SuperDARN data, specifically clarifying that we utilized the fitacf2.5 dataset. We have also included a citation for the software toolkit that produces this dataset, allowing readers to access detailed information on the data processing procedures and underlying assumptions. Furthermore, we have included a brief description with relevant citations on how SuperDARN datasets can be used to extract Medium-Scale Traveling Ionospheric Disturbances (MSTIDs) parameters. By incorporating these changes, we aim to give readers a clear understanding of the data processing and analytical steps involved in our study.*

  4. **Line 116:** What is SECS inversion technique? Why have you chosen this approach over the others? Details are needed to enhance the understanding of the reader.

***Reply.*** *This is now discussed in more detail in the revised Section 2.*

  5. **Line 123-136:** Similar to other comments on the previous instruments.

6. **Line 134-135:** The authors need to give enough details on this procedure instead of citing reference.

*Reply.* *More details are provided.*

7. **Line 138-142:** This section is not supposed to be here. It should be in the acknowledgement.

**Reply.** *While this looks like Acknowledgement, it is one important data source used in this study.*

*Revised and included as subsection 2.6. Solar wind data*

- **Result**

  o **AGWs/TIDs originating from lower thermosphere at high latitudes**

1. **Line 146-167:** This section needs to be in the introduction.

**Reply.** *Moved to the introduction.*

◻ **Event of January 8/9, 2013** The presentation

of this section:

2. the presentation is not arrange such that the reader can easily understand.

**Reply.** *Yes, the introduction to this section confounded by too many references may have been confusing. This is now clarified, the section content revised, and we believe better understandable to the reader.*

3. the presentation of the corresponding Figures is not in sequential order. For instance, in some part of the text, results presented in Figure 9 are presented before Figure 8.

**Reply.** *Figures have several panels, some of which are referred to later in the text, but we have rearranged the text and reference the figures in sequential order.*

4. some undefined abbreviations are found within this section.

**Reply.** *We make sure that the abbreviations have been explained.*

◻ **Events of November 1 and 4-5, 2014**

5.  What are the parameters of the observed waves? How do you know they are large-scale characteristics? This section needs to be revised. Either a table or a plot needs to be provided with the wave parameters. Putting some of them in the texts is not enough.

*Reply. The ground-scatter range mapping (Bristow, Greenwald and Samson, 1994; Frissell et al., 2014) is now applied to estimate the characteristics of the observed waves. In Section 3.1, Figure 3d have been revised, and similarly in Section 3.2 the estimated wave characteristics are discussed, and figures are provided in the Supplement.*

1.  **Line 301-302:** *Atlantic are sources of the GWs, which supports previously published results referenced above and points to winter jet stream as a likely source of GWs.*

    a. Please cite some references.

*Reply. A reference is now cited in the introduction of Section 4.*

**◻ Events of November 1-8, 2014**

2.  **Line 376-378: …** *were likely sources of MSTIDs propagating eastward to southeastward, as observed in the detrended vTEC maps (indicated by arrows in Figs. 13a,b) on November 1 and 8, 2014.*

    a. It has been mentioned that the propagation of the MSTIDs are indicated by arrows in Figure 13a.b. However, no arrows have been plotted. Only ">>" were used. Kindly use real arrows.

*Reply. The arrows have been edited.*

3.  **Line 388-390:** *As described in more detail by Chum and Podolská (2018) and Chum et al. (2021), the use of well correlated signals at two or three different frequencies makes it possible to determine a 3-D phase velocity vector.*

    a. How will the reader know the details in Chum and Podolská (2018) and Chum et al. (2021) with respect to the obtained result? Kindly mention here the exact point of these references.

*Reply. The description of the HF Doppler sounder technique along with these references are now part of Section 2.2. Section 4 has been significantly revised and a new, more recent event, with better GNSS coverage is added.*

**◻ Physical mechanism of GW generation in the troposphere**

4. **Line 439:** *Using the ERA5 reanalysis, similar to Figs. 15e,f, north-eastward propagating GWs in the ….*

      a. This is a bit confusing. There is no Figs. 15e,f. Kindly check and correct.

*Reply.* *This is now corrected in Section 4.4 discussing Fig. 17.*

• **Discussion**

1. **Line 454-455:** "*In Section 3.1, we have shown evidence that even during a geomagnetically very quiet period the TIDs that were observed by PFISR in Alaska can be attributed to sources at high latitudes*".

      a. This aspect of the manuscript in Section 3.1, appear more of literature review and presentation of result. I would like to encourage the authors to really show (with diagrams) other evidence that the high latitude sources were really the possible sources of the detected TIDs.

*Reply.* *As we have already explained in a reply above, all TID cases presented in Section 3.1 and 3.2 are shown to have sources at high latitudes. The Joule heating is a primary energy source for the night-time TIDs triggered in the auroral region, while the daytime TIDs can be also driven by precipitating particles in the polar cusp. PIFs are known to be associated with poleward moving auroral forms (precipitation), a source of AGWs/TIDs.*

**Reply to Referee Report 2**

**Title:** Observations of traveling ionospheric disturbances driven by gravity waves from sources in the upper and lower atmosphere.

**Overview of Manuscript**

This work presents Observations of traveling ionospheric disturbances driven by gravity waves from sources in the upper and lower atmosphere. They use a multi-instrument approach with the aim of attributing observed TIDs to atmospheric gravity waves generated in the lower thermosphere at midlatitudes. The work has the potential to contribute to existing literature if revised and some issues fixed.

I therefore recommend the manuscript be accepted after revision.

**Comments**

1.  The authors did not capture the important aspects of the work done. Please rewrite the abstract so it captures the attention of the reader and gives a clear overview of the work.

    *Reply 1. We have rewritten the abstract to better capture the aspects of this work, highlighting the multi-instrument techniques of TIDs observations and physical mechanisms that are sources of atmospheric gravity waves driving the TIDs.*

2.  The introduction is not very clear. It is very confusing to read and is more of a literature review than an introduction of the subject matter. Please rewrite it.

    *Reply 2.  The introduction has now been rewritten and reorganized. In addition to a literature review that is mandatory, the content of the manuscript is introduced. The purpose of this work to show that gravity waves driving equatorward propagating TIDs at high latitudes are generated by solar wind – MIT coupling on the dayside, even during low geomagnetic activity. The eastward propagating TIDs waves are attributed to sources of gravity waves generated in the troposphere by geostrophic adjustment processes and shear instability. This physical mechanism has not been previously considered as a source of TIDs.*

3.  The authors keep citing and referring the reader to works done without stating exactly in the text what they want the reader to know. Please cite and state the point you want to let the readers know in the text.

    *Reply 3.  In the introduction, the referencing of the published papers is now organised in paragraphs, each dedicated to specific aspects of the previous work (i.e., theory of GWs and their*

*relation to aurorally generated TIDs, lower atmospheric sources of GWs/TIDs (including polar vortex that has been recently, in our opinion, overemphasized), and TIDs detection techniques.*

4. Please elaborate on the SECS inversion technique and it´s advantage over other techniques.

   ***Reply 4.*** *Descriptions of the techniques, including the SECS inversion technique, are now expanded and reorganized into subsections of Section 2.*

5. Please can the authors represent the instruments used and also the location of the studies in a table or map? This will help the reader.

   ***Reply 5.*** *A map of instruments used in this study is now shown in Figure 1.*

6. The methods used to arrive at the results are very shallow and would be difficult for replication by the reader. The authors should please give detailed and step-by-step writeup of the methodology employed.

   ***Reply 6.*** *In addition to rewritten and expanded Section 2, Sections 3 and 4 have been significantly revised discussing specific results in more detail.*

7. Please the authors should state clearly in the text the contribution (the new findings) of this work to already existing literature.

***Reply 7.*** *We have now more clearly stated the contribution of this work in Sections 1, 5 and 6 (i.e., solar wind – MIT coupling on the dayside generating TIDs, even during low geomagnetic activity; sources of gravity waves generated in the troposphere by geostrophic adjustment processes driving TID*

---

## Author Response (AR3)

**Reply to Referee Report 1**

**Title:** Observations of traveling ionospheric disturbances driven by gravity waves from sources in the upper and lower atmosphere.

**Overview of Manuscript**

This work investigates the sources of ionospheric disturbances due to atmospheric gravity waves due to high latitude space weather and low latitude tropospheric dynamics. They used multiple observational techniques to detect traveling ionospheric disturbances (TIDs) with large and medium scale characteristics, and atmospheric gravity waves (AGWs). Using observations and reanalysis data, they investigated the upper atmospheric dynamics that possibly excited the large and medium scale TIDs originating from the troposphere. The manuscript has the potential to contribute significantly to literature; however, these minor issues need to be addressed.

The authors have responded well to comments and implemented satisfactorily my comments and suggestions. However, there are some minor issues that needs to be solved, especially in the abstract and data sources and methods sections. In the case of the abstract, I will recommend that the authors consider rewritting it entirely with concise but direct and needed details.

The manuscript in the current needs minor revision after which it can be accepted for publication.

**Abstract:**

**General Comment**

The abstract is quite confusing. This, I consider to come from the presentation of instruments used and their respective methodologies. Please, carefully but in simple terms consider writting the instrumentation and methodology aspects. Also, capture in clear terms the major contribution of the work.

*Reply. We have rewritten the abstract in simple terms to capture the major contributions of this work.*

**Minor Comment(s):**

1. **Lines 22 and 24:** Kindly consider rephrasing the sentence: " .... generate gravity waves driving equatorward propagating medium- to large-scale traveling ionospheric disturbances (TIDs) ... and ..... extratropical cyclones are sources of gravity waves driving medium-scale TIDs. The phrase "generate gravity waves driving" .... is a bit confusing.
*Reply. The sentence has been now rephrased.*

**Data sources and methods**

**General Comment(s):**

The authors have implemented satisfactorily in relation to this section. However, I would like to suggest to them to iclude the map of the GNSS reciever stations in the supplementary materials to the Fig.1 here (in the main manuscript). Thus, making Fig.1 to comprise of panels (a) – for the current fig. 1, (b) – for the first local domain, and (c) – for the second local domain. The legends are not explained in the text. Infact no legend were defined in the figures for the maps (both the one in the main manuscipt and in the supplementary material). Kindly update them and explain them in the text according to the paragraph in which each instrument was described.

*Reply: We have revised Figure 1 by including the legends and merging it with Figure S1 from the supplementary materials. It is now referenced and explained in the text where each instrument is described.*

**Minor Comment:**

1. Line 24: The GNSS data for this ???? . .... The sentence seems incomplete, kindly revise.

*Reply: This incomplete sentence is now revised. Thank you.*

**Reply to Referee Report 2**

**Title:** Observations of traveling ionospheric disturbances driven by gravity waves from sources in the upper and lower atmosphere.

**Overview of Manuscript**

This work presents Observations of traveling ionospheric disturbances driven by gravity waves from sources in the upper and lower atmosphere. They use a multi-instrument approach with the aim of attributing observed TIDs to atmospheric gravity waves generated in the lower thermosphere at midlatitudes. The work has the potential to contribute to existing literature if revised and some issues fixed.

I therefore recommend the manuscript be accepted after the implementation of the comment.

**Comments**

1. The authors should please further simplify the abstract capturing only the important aspects. Although they have worked on it considering the previous version, they should be encouraged to go straight to the point.

*Reply: The abstract has been now simplified to capture the important aspects.*

*We would like to thank both reviewers for helping us to improve this manuscript.*